# A Communication-Efficient Distributed Gradient Clipping Algorithm for Training Deep Neural Networks

## Abstract

In distributed training of deep neural networks or Federated Learning (FL), people usually run Stochastic Gradient Descent (SGD) or its variants on each machine and communicate with other machines periodically. However, SGD might converge slowly in training some deep neural networks (e.g., RNN, LSTM) because of the exploding gradient issue. Gradient clipping is usually employed to address this issue in the single machine setting, but exploring this technique in the FL setting is still in its infancy: it remains mysterious whether the gradient clipping scheme can take advantage of multiple machines to enjoy parallel speedup in the FL setting. The main technical difficulty lies at dealing with nonconvex loss function, non-Lipschitz continuous gradient, and skipping communication rounds simultaneously. In this paper, we explore a relaxed-smoothness assumption of the loss landscape which LSTM was shown to satisfy in previous works, and design a communication-efficient gradient clipping algorithm. This algorithm can be run on multiple machines, where each machine employs a gradient clipping scheme and communicate with other machines after multiple steps of gradient-based updates. Our algorithm is proved to have $O\left(\frac{1}{N\epsilon^4}\right)$ iteration complexity for finding an $\epsilon$-stationary point, where $N$ is the number of machines. This indicates that our algorithm enjoys linear speedup. Our experiments on several benchmark datasets and various scenarios demonstrate that our algorithm indeed exhibits fast convergence speed in practice and validate our theory.

## 1 Introduction

Deep learning has achieved tremendous successes in many domains including computer vision (Krizhevsky et al., 2012; He et al., 2016) and natural language processing (Devlin et al., 2018), game (Silver et al., 2016). To obtain good empirical performance, people usually need to train large models on a huge amount of data, and it is usually very computationally expensive. To speedup the training process, distributed training becomes indispensable (Dean et al., 2012). For example, Goyal et al. (2017) trained a ResNet-50 on ImageNet dataset by distributed SGD with minibatch size 8192 on 256 GPUs in only one hour, which not only matches the small minibatch accuracy but also enjoys parallel speedup, and hence improves the running time. Recently, there is an increasing interest in an variant of distributed learning, namely Federated Learning (FL) (McMahan et al., 2017), which focuses on the cases where the training data is non-i.i.d. across devices and only limited communication is allowed. McMahan et al. (2017) proposed an algorithm named Federated Averaging, which runs multiple steps of SGD on each clients before communicating with other clients.

Despite the empirical success of distributed SGD and its variants (e.g., Federated Averaging) in deep learning, they may not exhibit good performance when training some neural networks (e.g., Recurrent Neural Networks, LSTMs), due to the exploding gradient problem (Pascanu et al., 2012; 2013). To address this issue, Pascanu et al. (2013) proposed to use the gradient clipping strategy, and it has become a standard technique when training language models (Gehring et al., 2017; Peters et al., 2018; Merity et al., 2018). There are some recent works trying to theoretically explain gradient clipping from nonconvex optimization's perspective (Zhang et al., 2019; 2020). These works are built upon an important observation made in (Zhang et al., 2019): for certain neural networks such as LSTM, the gradient does not vary uniformly over the loss landscape (i.e., the gradient is not Lipschitz continuous with a uniform constant), and the gradient Lipschitz constant can scale linearly with respect to the gradient norm. This is referred to as the relaxed smoothness condition (i.e., $(L_0, L_1)$-smoothness defined in Definition 2), which generalizes but strictly relaxes the usual smoothness condition (i.e., $L$-smoothness defined in Definition 1). Under the relaxed smoothness

Table 1: Comparison of Iteration and Communication Complexity of Different Algorithms for finding a point whose gradient's magnitude is smaller than $\epsilon$ (i.e., $\epsilon$-stationary point defined in Definition 3), $N$ is the number of machines, the meaning of other constants can be found in Assumption 1.

| Algorithm | Setting | Iteration Complexity | Communication Complexity |
|---|---|---|---|
| SGD (Ghadimi & Lan, 2013) | Single[1] | $O\left(\Delta(L_0 + L_1 M)\epsilon^{-2} + \Delta(L_0 + L_1 M)\sigma^2\epsilon^{-4}\right)$ | N/A |
| Clipped SGD (Zhang et al., 2019) | Single | $O\left((\Delta + (L_0 + L_1\sigma)\sigma^2 + \sigma L_0^2/L_1)^2)\epsilon^{-4}\right)$ | N/A |
| Clipping Framework (Zhang et al., 2020) | Single | $O\left(\Delta L_0 \sigma^2 \epsilon^{-4}\right)$ | N/A |
| Naive Parallel of (Zhang et al., 2020) | Distributed[2] | $O\left(\Delta L_0 \sigma^2/(N\epsilon^4)\right)$ | $O\left(\Delta L_0 \sigma^2/(N\epsilon^4)\right)$ |
| Ours (this work) | FL | $O(\Delta L_0(\sigma + \kappa)^2/(N\epsilon^4))$ | $O\left(\Delta L_0(\sigma + \kappa)^2 \epsilon^{-3}\right)$ |

condition, Zhang et al. (2019; 2020) proved that gradient clipping enjoys polynomial-time iteration complexity for finding the first-order stationary point in the single machine setting, and it can be arbitrarily faster than fix-step gradient descent.

In practice, both distributed learning (or FL) and gradient clipping are important techniques to accelerate neural network training. However, the theoretical analysis of gradient clipping is only restricted to the single machine setting (Zhang et al., 2019; 2020). Hence it naturally motivates us to consider the following question:

**Is it possible that the gradient clipping scheme can take advantage of multiple machines to enjoy parallel speedup in training deep neural networks, with data heterogeneity across machines and limited communication?**

In this paper, we give an affirmative answer to the above question. Built upon the relaxed smoothness condition as in (Zhang et al., 2019; 2020), we design a communication-efficient distributed gradient clipping algorithm. The key characteristics of our algorithm are: (i) unlike naive parallel gradient clipping algorithm which requires averaging model weights and gradients from all machines for every iteration, our algorithm only aggregates weights with other machines after certain number of local updates on each machine; (ii) our algorithm clips the gradient according to the norm of the local gradient on each machine, instead of the norm of the averaged gradients across machines as in the naive parallel version. These key features make our algorithm amenable to the FL setting and it is nontrivial to establish desired theoretical guarantees (e.g., linear speedup, reduced communication complexity). The main difficulty in the analysis lies at dealing with nonconvex objective function, non-Lipschitz continuous gradient, and skipping communication rounds simultaneously. Our main contribution is summarized as the following:

- We design a novel communication-efficient distributed stochastic local gradient clipping algorithm, namely CELGC, for solving a nonconvex optimization problem under the relaxed smoothness condition. The algorithm only needs to clip the gradient according to the local gradient's magnitude and globally averages the weights on all machines periodically. To the best of our knowledge, this is the first work proposing communication-efficient distributed stochastic gradient clipping algorithms under the relaxed smoothness condition.

- Under the relaxed smoothness condition, we prove iteration and communication complexity results of our algorithm for finding an $\epsilon$-stationary point. First, comparing with (Zhang et al., 2020), we prove that our algorithm enjoys linear speedup, which means that the iteration complexity of our algorithm is reduced by a factor of $N$ (the number of machines). Second, comparing with naive parallel verion of the algorithm of (Zhang et al., 2020), we

---

[1]In this setting, we assume the gradient norm is upper bounded by $M$ such that the gradient is $(L_0 + L_1 M)$-Lipschitz. However, we want to emphasize that the original paper of (Ghadimi & Lan, 2013) does not require bounded gradient assumption, instead they require $L$-Lipschitz gradient and bounded variance $\sigma^2$. Under their assumption, their complexity result is $O\left(\Delta L\epsilon^{-2} + \Delta L\sigma^2\epsilon^{-4}\right)$.

[2]Naive Parallel of (Zhang et al. 2020) is different from our algorithm (CELGC) with $I = 1$ in that the naive version requires averaging gradients across all machines to update the model while CELGC only updates the model using local gradients computed in that machine. This also means that in each iteration, naive version clips the gradient based on the globally averaged gradient while ours only bases on the local gradient.

prove that our algorithm enjoys better communication complexity. Specifically, our algorithm's communication complexity is smaller than naive parallel clipping algorithm if the number of machines is not too large (i.e., $N \leq O(1/\epsilon)$). The detailed comparison over existing algorithms under the same relaxed smoothness condition is described in Table 1. Please refer to (Koloskova et al., 2020) for local SGD complexity results for $L$-smooth functions.

- We empirically verify our theoretical results by conducting experiments on different neural network architectures on benchmark datasets and on various scenarios including small to large batch-sizes, homogeneous and heterogeneous data distributions, and partial participation of machines. The experimental results demonstrate that our proposed algorithm indeed exhibit speedup in practice.

## 2 RELATED WORK

**Gradient Clipping/Normalization Algorithms**    In deep learning literature, gradient clipping (normalization) technique was initially proposed by (Pascanu et al., 2013) to address the issue of exploding gradient problem in (Pascanu et al., 2012), and it has become a standard technique when training language models (Gehring et al., 2017; Peters et al., 2018; Merity et al., 2018). Menon et al. (2019) showed that gradient clipping is robust and can mitigate label noise. Recently gradient normalization techniques (You et al., 2017; 2019) were applied to train deep neural networks on the very large batch setting. For example, You et al. (2017) designed LARS algorithm to train a ResNet50 on ImageNet with batch size 32k, which utilized different learning rate according to the norm of the weights and the norm of the gradient.

In optimization literature, gradient clipping (normalization) was used in early days in the field of convex optimization (Ermoliev, 1988; Alber et al., 1998; Shor, 2012). Nesterov (1984) and Hazan et al. (2015) considered normalized gradient descent for quasi-convex functions in deterministic and stochastic cases respectively. Gorbunov et al. (2020) designed an accelerated gradient clipping method to solve convex optimization problem with heavy-tailed noise in stochastic gradients. Mai & Johansson (2021) established the stability and convergence of stochastic gradient clipping algorithms for convex and weakly convex functions. In nonconvex optimization, Levy (2016) showed that normalized gradient descent can escape from saddle points. Cutkosky & Mehta (2020) showed that adding a momentum provably improves the normalized SGD in nonconvex optimization. Zhang et al. (2019) and Zhang et al. (2020) analyzed the gradient clipping for nonconvex optimization under the relaxed smoothness condition rather than the traditional $L$-smoothness condition in nonconvex optimization (Ghadimi & Lan, 2013).

However, all of them only consider the algorithm in the single machine setting or the naive parallel setting, and none of them can apply to FL setting where data on different nodes is heterogeneous and only limited communication is allowed.

**Communication-Efficient Algorithms in Distributed and Federated Learning**    In large-scale machine learning, people usually train their model using first-order methods on multiple machines and these machines communicate and aggregate their model parameters periodically. When the function is convex, there is scheme named one-shot averaging (Zinkevich et al., 2010; McDonald et al., 2010; Zhang et al., 2013; Shamir & Srebro, 2014), in which every machine runs an stochastic approximation algorithm and averages the model weights across machines only at the very last iteration. One-shot averaging scheme is communication-efficient and enjoys statistical convergence with one pass of the data (Zhang et al., 2013; Shamir & Srebro, 2014; Jain et al., 2017; Koloskova et al., 2019), but the training error may not converge in practice. McMahan et al. (2017) considered the Federated Learning setting where the data is decentralized and might be non-i.i.d. across devices and communication is expensive. McMahan et al. (2017) designed the very first algorithm for FL (a.k.a., FedAvg), which is communication-efficient since every node communicates with other nodes infrequently. Stich (2018) considered a concrete case of FedAvg, namely local SGD, which runs SGD independently in parallel on different works and averages the model parameters only once in a while. Stich (2018) also showed that local SGD enjoys linear speedup for strongly-convex objective function. There are also some works analyzing local SGD and its variants on convex (Dieuleveut & Patel, 2019; Khaled et al., 2020; Karimireddy et al., 2020; Woodworth et al., 2020a;b; Gorbunov et al., 2021; Yuan et al., 2021) and nonconvex smooth functions (Zhou & Cong, 2017; Yu et al., 2019a;b; Jiang & Agrawal, 2018; Wang & Joshi, 2018; Lin et al., 2018; Basu et al.,

2019; Haddadpour et al., 2019; Karimireddy et al., 2020). Recently, Woodworth et al. (2020a;b) analyzed advantages and drawbacks of local SGD compared with minibatch SGD for convex objectives. Woodworth et al. (2021) proved hardness results for distributed stochastic convex optimization. Reddi et al. (2021) introduced a general framework of federated optimization and designed several federated versions of adaptive optimizers. Zhang et al. (2021) considered to employ gradient clipping to optimize $L$-smooth functions and achieve differential privacy. Due to a vast amount of literature of FL and limited space, we refer readers to (Kairouz et al., 2019) and references therein.

However, all of these works either assume the objective function is convex or $L$-smooth. To the best of our knowledge, our algorithm is the first communication-efficient algorithm which does not rely on these assumptions but still enjoys linear speedup.

## 3 PRELIMINARIES, NOTATIONS AND PROBLEM SETUP

**Preliminaries and Notations**  Denote $\|\cdot\|$ by the Euclidean norm. We denote $f : \mathbb{R}^d \to \mathbb{R}$ as the overall loss function, and $f_i : \mathbb{R}^d \to \mathbb{R}$ as the loss function on $i$-th machine, where $i = 1, \ldots, N$. Denote $\nabla h(\mathbf{x})$ as the gradient of $h$ evaluated at the point $\mathbf{x}$, and denote $\nabla h(\mathbf{x}; \xi)$ as the stochastic gradient of $h$ calculated based on sample $\xi$.

**Definition 1** ($L$-smoothness). *A function $h$ is $L$-smooth if $\|\nabla h(\mathbf{x}) - \nabla h(\mathbf{y})\| \leq L\|\mathbf{x} - \mathbf{y}\|$ for all $\mathbf{x}, \mathbf{y} \in \mathbb{R}^d$.*

**Definition 2** (($L_0, L_1$)-smoothness). *A second order differentiable function $h$ is $(L_0, L_1)$-smooth if $\|\nabla^2 h(\mathbf{x})\| \leq L_0 + L_1\|\nabla h(\mathbf{x})\|$ for any $\mathbf{x} \in \mathbb{R}^d$.*

**Definition 3** ($\epsilon$-stationary point). *$\mathbf{x} \in \mathbb{R}^d$ is an $\epsilon$-stationary point of the function $h$ if $\|\nabla h(\mathbf{x})\| \leq \epsilon$.*

**Remark:**  From definitions, we know that the $(L_0, L_1)$-smoothness is strictly weaker than $L$-smoothness. To see this, first, we know that $L$-smooth functions is $(L_0, L_1)$-smooth with $L_0 = L$ and $L_1 = 0$. However the reverse is not true. For example, consider the function $h(x) = x^4$, we know that the gradient is not Lipschitz continuous and hence is not $L$-smooth, but $|h''(x)| = 12x^2 \leq 12 + 3 \times 4|x|^3 = 12 + 3|h'(x)|$, so $h(x) = x^4$ is $(12, 3)$-smooth. Zhang et al. (2019) empirically showed that the $(L_0, L_1)$-smoothness holds for the AWD-LSTM (Merity et al., 2018). In nonconvex optimization literature (Ghadimi & Lan, 2013; Zhang et al., 2020), the goal is to find an $\epsilon$-stationary point since it is NP-hard to find a global optimal solution for a general nonconvex function.

**Problem Setup**  In this paper, we consider the following optimization problem:

$$\min_{\mathbf{x} \in \mathbb{R}^d} f(\mathbf{x}) = \frac{1}{N} \sum_{i=1}^{N} f_i(\mathbf{x}), \tag{1}$$

where $N$ is the number of nodes and each $f_i(\mathbf{x}) := \mathbb{E}_{\xi_i \sim \mathcal{D}_i}[F_i(\mathbf{x}; \xi_i)]$ is a nonconvex function where $\mathcal{D}_i$ can be possibly different for different $i$. This formulation has broad applications in distributed deep learning and FL. For example, in FL setting, $f_i$ stands for the loss function on $i$-th machine, $\mathcal{D}_i$ represents the data distribution on $i$-th machine, and $N$ machines want to jointly optimize the objective function $f$.

We make the following assumptions throughout the paper.

**Assumption 1.**  *(i) Each function $f_i(\mathbf{x})$ is $(L_0, L_1)$-smooth, i.e., $\|\nabla^2 f_i(\mathbf{x})\| \leq L_0 + L_1\|\nabla f_i(\mathbf{x})\|$, for $\forall \mathbf{x} \in \mathbb{R}^d$ and $i = 1, \ldots, N$.*

*(ii) There exists $\Delta > 0$ such that $f(\mathbf{x}_0) - f_* \leq \Delta$, where $f_*$ is the global optimal value of $f$.*

*(iii) For all $\mathbf{x} \in \mathbb{R}^d$, $\mathbb{E}_{\xi_i \sim \mathcal{D}_i}[\nabla F_i(\mathbf{x}; \xi_i)] = \nabla f_i(\mathbf{x})$, and $\|\nabla F_i(\mathbf{x}; \xi) - \nabla f_i(\mathbf{x})\| \leq \sigma$ almost surely.*

*(iv) $\frac{1}{N} \sum_{i=1}^{N} \|\nabla f_i(\mathbf{x}) - \nabla f(\mathbf{x})\| \leq \kappa$.*

**Remark:**  The Assumption 1 (i) means that that the loss function defined on each machine satisfies the relaxed-smoothness condition, and it holds when we want to train a language model using LSTMs. Assumption 1 (ii) and (iii) are standard assumptions in nonconvex optimization (Ghadimi & Lan, 2013; Zhang et al., 2019). Note that it is usually assumed that the stochastic gradient is unbiased and has bounded variance (Ghadimi & Lan, 2013), but we follow (Zhang et al., 2019) to assume we have unbiased stochastic gradient with almost surely bounded deviation $\sigma$. This is an

---

**Algorithm 1** Communication Efficient Local Gradient Clipping (CELGC)

---

1: **for** $t = 0, \ldots, T$ **do**
2:     Each node $i$ samples its stochastic gradient $\nabla F_i(\mathbf{x}_t^i; \xi_t^i)$, where $\xi_t^i \sim \mathcal{D}_i$.
3:     Each node $i$ updates it local solution **in parallel**:

$$\mathbf{x}_{t+1}^i = \mathbf{x}_t^i - \min\left(\eta, \frac{\gamma}{\|\nabla F_i(\mathbf{x}_t^i; \xi_t^i)\|}\right) \nabla F_i(\mathbf{x}_t^i; \xi_t^i) \tag{2}$$

4:     **if** $t$ is a multiple of $I$ **then**
5:         Each worker resets the local solution as the averaged solution across nodes:

$$\mathbf{x}_t^i = \widehat{\mathbf{x}} := \frac{1}{N} \sum_{j=1}^{N} \mathbf{x}_t^j \qquad \forall i \in \{1, \ldots, N\} \tag{3}$$

6:     **end if**
7: **end for**

---

stronger assumption than the bounded variance, but it is a normal assumption when encountering relaxed smoothness. Assumption 1 (iv) quantifies the averaged heterogeneity across nodes, which is frequently used in the FL literature (e.g., Yu et al. (2019a)).

## 4 ALGORITHM AND THEORETICAL ANALYSIS

### 4.1 MAIN DIFFICULTY AND THE ALGORITHM DESIGN

We briefly present the main difficulty in extending the single machine setting (Zhang et al., 2020) to the FL setting. In (Zhang et al., 2020), they split the contribution of decreasing objective value by considering two cases: clipping large gradients and keeping small gradients. If communication is allowed at every iteration, then we can aggregate gradients on each machine and determine whether we should clip or keep the averaged gradient or not. However, in FL setting, communicating with other machines at every iteration is not allowed. This would lead to the following difficulties: (i) the averaged gradient may not be available to the algorithm if communication is limited, so it is hard to determine whether clipping operation should be performed or not; (ii) the model weight on every machine may not be the same when communication does not happen at the current iteration; (iii) the loss function is not $L$-smooth, so the usual local SGD analysis for $L$-smooth functions cannot be applied in this case.

To address this issue, we design a new algorithm, namely Communication-Efficient Local Gradient Clipping (CELGC), which is presented in Algorithm 1. The algorithm calculates a stochastic gradient and then performs multiple local gradient clipping steps on each machine in parallel, and aggregates model parameters on all machines after every $I$ steps of local updates. Note that the naive version of the parallel gradient clipping algorithm in (Zhang et al., 2020) needs to aggregates model parameters and gradients across all machines at every iteration, and perform one step of gradient clipping operation based on the aggregated gradient. Conceptually speaking, our algorithm is expected to have better performance. The reason is that our algorithm is able to skip communication rounds, and does not need to transmit gradient information across machines (note that it only averages the weights). The remaining issue is that the non-asymptotic convergence guarantees are not established yet. In other words, we aim to establish iteration and communication complexity for Algorithm 1 for finding an $\epsilon$-stationary point. We present our main theoretical results as below.

### 4.2 MAIN RESULTS

**Theorem 1.** *Suppose Assumption 1 holds and $\sigma + \kappa \geq 1$. Take $\epsilon \leq \min(\frac{AL_0}{BL_1}, 0.1)$ be a small enough constant and $N \leq \min(\frac{1}{\epsilon}, \frac{14AL_0}{5BL_1\epsilon})$. In Algorithm 1, choose $I \leq \frac{1}{2N\epsilon}$, $\gamma \leq \frac{N\epsilon}{28(\sigma+\kappa)} \min\{\frac{\epsilon}{AL_0}, \frac{1}{BL_1}\}$ and the fixed ratio $\frac{\gamma}{\eta} = 5(\sigma + \kappa)$, where $A \geq 1$ and $B \geq 1$ are constants which will be specified in the proof, and run Algorithm 1 for $T = O\left(\frac{\Delta L_0}{N\epsilon^4}\right)$ iterations. Then we have*

$$\frac{1}{T} \sum_{t=1}^{T} \mathbb{E}\|\nabla f(\bar{\mathbf{x}}_t)\| \leq 4\epsilon.$$

**Remark:** We have some implications of Theorem 1. When the number of machines is not large (i.e., $N \leq O(1/\epsilon)$) and the number of skipped communications is not large (i.e., $I \leq O(1/\epsilon N)$), then with proper setting of learning rate, we have following observations. First, our algorithm enjoys linear speedup, since the number of iterations we need to find an $\epsilon$-stationary point is divided by the number of machines $N$ when comparing the single machine algorithm in (Zhang et al., 2020). Second, our algorithm is communication-efficient, since the communication complexity is $T/I = O\left(\Delta L_0 (\sigma + \kappa)^2 \epsilon^{-3}\right)$, which provably improves the naive parallel gradient clipping algorithm of (Zhang et al., 2020) with $O(\Delta L_0 \sigma^2 / (N \epsilon^4))$ communication complexity when $N \leq O(1/\epsilon)$.

Another important fact is that both iteration complexity and communication complexity only depend on $L_0$, independent of $L_1$ and the gradient upper bound $M$. This indicates that our algorithm does not suffer from slow convergence even if these quantities are large. In addition, local gradient clipping is a good mechanism to alleviate the bad effects brought by a rapidly changing loss landscape (e.g., some language models such as LSTM).

### 4.3 SKETCH OF THE PROOF OF THEOREM 1

In this section, we present the sketch of our proof of Theorem 1 and the detailed proof can be found in Appendix B. The key idea in our proof is to establish the descent property of the sequence $\{f(\bar{\mathbf{x}}_t)\}_{t=0}^T$ in the FL setting under the relaxed smoothness condition, where $\bar{\mathbf{x}}_t = \frac{1}{N} \sum_{i=1}^t \mathbf{x}_t^i$ is the averaged weight across all machines at $t$-th iteration. The main challenge is that the descent property of $(L_0, L_1)$-smooth function in the FL setting does not naturally hold, which is in sharp contrast to the usual local SGD proof for $L$-smooth functions. To address this challenge, we need to carefully study whether the algorithm is able to decrease the objective function in different situations. Our main technical innovations in the proof are listed as the following.

First, we monitor the algorithm's progress in decreasing the objective value according to some novel measures. The measures we use are the magnitude of the gradient evaluated at the averaged weight and the magnitude of local gradients evaluated at the individual weights on every machine. Please note that our algorithm does not have access to the gradient evaluated at the averaged weight, but it can be served as a proxy in our proof even if we do not have knowledge about it. To this end, we introduce Lemma 2, whose goal is to carefully inspect how much progress the algorithm makes, according to the magnitude of local gradients calculated on each machine. The reason is that the local gradient's magnitude is an indicator of whether the clipping operation happens or not. For each fixed iteration $t$, we define $J(t) = \{i \in \{1 : N\} : \|\nabla F_i(\mathbf{x}_t^i, \xi_t^i)\| \geq \gamma/\eta\}$ and $\bar{J}(t) = \{1 : N\} \backslash J(t)$. Briefly speaking, $J(t)$ contains all machines that perform clipping operation at iteration $t$ and $\bar{J}(t)$ is the set of machines that do not perform clip operation at iteration $t$. In Lemma 2, we perform one-step analysis and consider all machines with different clipping behaviors at the iteration $t$. By considering all cases together and taking the telescoping sum over $t = 0, \ldots, T$, we can get an upper bound of the gradient in the ergodic sense.

Second, Zhang et al. (2020) inspect their algorithm's progress by considering the magnitude of gradient at different iterations, so they treat every iteration differently. However, this approach does not work in FL setting since one cannot get access to the averaged gradient across machines at every iteration. Instead, we treat every iteration of the algorithm as the same but consider the progress made by each machine.

Third, by properly choosing hyperparameters $(\eta, \gamma, I)$ and using an amortized analysis, we prove that our algorithm can decrease the objective value by an sufficient amount, and the sufficient decrease is mainly due to the case where the gradient is not too large (i.e., clipping operations do not happen). This important insight allows us to better characterize the training dynamics without worrying too much about the case that where gradient is large(i.e., the clipping operation is performed).

With the idea mentioned above, now we present how to proceed with the proof in detail.

Lemma 1 characterizes the $\ell_2$ error between averaged weight and individual weights at $t$-th iteration. Intuitively, the $\ell_2$ error scales linearly in terms of the length of node synchronization interval $I$.

**Lemma 1.** *Under Assumption 1, for any $i$ and $t$, Algorithm 1 ensures $\|\bar{\mathbf{x}}_t - \mathbf{x}_t^i\| \leq 2\gamma I$ holds almost surely.*

Lemma 4 and Lemma 5 (included in Appendix A) are some properties of $(L_0, L_1)$-smooth functions and we need to use them frequently in our paper. To make sure they work, we need $2\gamma I \leq c/L_1$ for

some $c > 0$. This inequality follows from the choice of parameters in Theorem 1 and details will be shown in Appendix B. We denote $A = 1 + e^c - \frac{e^c-1}{c}$ and $B = \frac{e^c-1}{c}$.

Let $J(t)$ be the index set of $i$ such that $\|\nabla F_i(\mathbf{x}_t^i, \xi_t^i)\| \geq \frac{\gamma}{\eta}$ at fixed iteration $t$, i.e., $J(t) = \{i \in [1, \ldots, N] \mid \|\nabla F_i(\mathbf{x}_t^i, \xi_t^i)\| \geq \frac{\gamma}{\eta}\}$. Lemma 2 characterizes how much progress we can get in one iteration of Algorithm 1, and the progress is decomposed into contributions from every machine (note that $J(t) \cup \bar{J}(t) = \{1, \ldots, N\}$ for every $t$).

**Lemma 2.** *Let $J(t)$ be the set defined as above. If $2\gamma I \leq c/L_1$ for some $c > 0$, If $AL_0\eta \leq 1/2$, then we have*

$$\mathbb{E}[f(\bar{\mathbf{x}}_{t+1}) - f(\bar{\mathbf{x}}_t)]$$

$$\leq \frac{1}{N}\mathbb{E}\sum_{i \in J(t)}\left[-\frac{2\gamma}{5}\|\nabla f(\bar{\mathbf{x}}_t)\| - \frac{3\gamma^2}{5\eta} + \frac{7\gamma}{5}\|\nabla F_i(\mathbf{x}_t^i; \xi_t^i) - \nabla f(\bar{\mathbf{x}}_t)\| + AL_0\gamma^2 + \frac{BL_1\gamma^2\|\nabla f(\bar{\mathbf{x}}_t)\|}{2} + \frac{AL_0\eta^2\sigma^2}{N}\right]$$

$$+\mathbb{E}\frac{1}{N}\sum_{i \in \bar{J}(t)}\left[-\frac{\eta}{2}\|\nabla f(\bar{\mathbf{x}}_t)\|^2 + 4\gamma^2 I^2 A^2 L_0^2\eta + 4\gamma^2 I^2 B^2 L_1^2\eta\|\nabla f(\bar{\mathbf{x}}_t)\|^2 + \frac{AL_0\eta^2\sigma^2}{N} + \frac{BL_1\gamma^2\|\nabla f(\bar{\mathbf{x}}_t)\|}{2}\right],$$

*where $A = 1 + e^c - \frac{e^c-1}{c}$ and $B = \frac{e^c-1}{c}$.*

Lemma 3 quantifies an upper bound of the averaged $\ell_2$ error between local gradient evaluated at the local weight and the gradient evaluated at the averaged weight. The upper bound contains the noise term in stochastic gradient $\sigma$, the data heterogeneity $\kappa$, and another error term which scales linearly with the length of node synchronization interval $I$.

**Lemma 3.** *Suppose Assumption 1 holds. If $2\gamma I \leq c/L_1$ for some $c > 0$, then we obtain*

$$\frac{1}{N}\sum_{i=1}^{N}\left\|\nabla F_i(\mathbf{x}_t^i; \xi_t^i) - \nabla f(\bar{\mathbf{x}}_t)\right\| \leq \sigma + \kappa + 2\gamma I(AL_0 + BL_1\|\nabla f(\bar{\mathbf{x}}_t)\|) \quad \text{almost surely,}$$

*where $A = 1 + e^c - \frac{e^c-1}{c}$ and $B = \frac{e^c-1}{c}$.*

**Putting all together** Suppose our algorithm runs $T$ iterations. Taking summation on both sides of Lemma 2 over all $t = 0, \ldots, T - 1$, we are able to get an upper bound of $\sum_{t=0}^{T-1}\mathbb{E}[f(\bar{\mathbf{x}}_{t+1}) - f(\bar{\mathbf{x}}_t)] = \mathbb{E}[f(\bar{\mathbf{x}}_T) - f(\bar{\mathbf{x}}_0)]$. Note that $\mathbb{E}[f(\bar{\mathbf{x}}_T) - f(\bar{\mathbf{x}}_0)] \geq -\Delta$ due to Assumption 1, so we are able to get a upper bound of gradient norm. For details, please refer to the proof of Theorem 1 in Appendix B.

## 5 EXPERIMENTS

We conduct extensive experiments to validate the merits of our algorithm in realistic settings and find the distributed clipping algorithm indeed consistently exhibits substantial speedup compared with the baseline, which is the naive parallel version of the algorithm in (Zhang et al., 2020). We want to re-emphasize that the major difference is that the baseline algorithm needs to average the model weights and local gradients at every iteration while ours only requires averaging the model weights after $I$ iterations and does not need to average the gradients at all. This immediately suggests that our algorithm will gain substantial speedup in terms of the wall clock time, which is also supported by our empirical experiments in this section.

We conduct each experiment in two nodes with 4 Nvidia-V100 GPUs for each node. In our experiments, one "machine" corresponds to one GPU, and we use the word "GPU" and "machine" in this section interchangeably. We compared our algorithm with the baseline across three deep learning benchmarks: CIFAR-10 image classification with ResNet, Penn Treebank language modeling with LSTM, and Wikitext-2 language modeling with LSTM. All algorithms and the the training framework are implemented in Pytorch 1.4. Due to limited computational resources, we choose the same hyperparameters (learning rates, clipping thresholds) according to the best-tuned baselines unless otherwise specified. For more results including small and large batch-sizes, heterogeneous data distribution, and partial participation of machines, we kindly refer readers to the Appendix C and E.

### 5.1 EFFECTS OF SKIPPING COMMUNICATION

We focus on one feature of our algorithm: skipping communication. Theorem 1 says that our algorithm enjoys reduced communication complexity since every node only communicates with

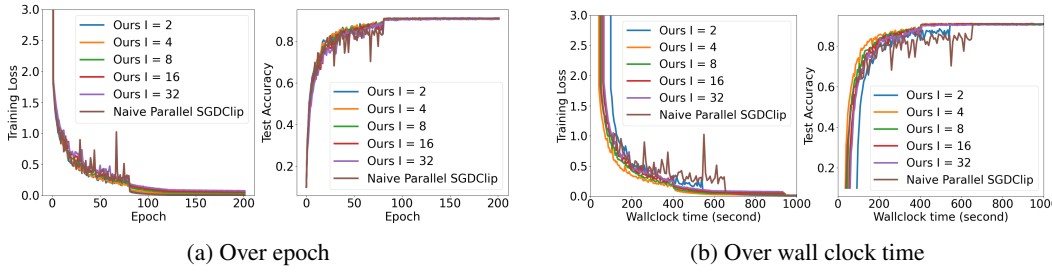

Figure 1: Algorithm 1 with different $I$: Training loss and test accuracy v.s. (Left) epoch and (right) wall clock time on training a 56 layer Resnet to do image classification on CIFAR10.

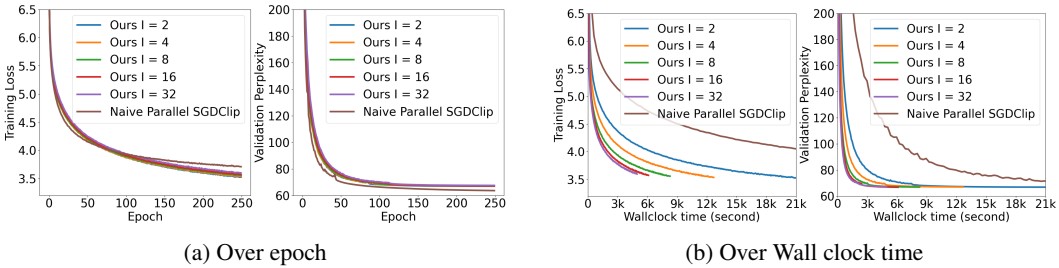

Figure 2: Algorithm 1 with different $I$: Training loss and validation perplexity v.s. (Left) epoch and (right) wall clock time on training an AWD-LSTM to do language modeling on Penn Treebank.

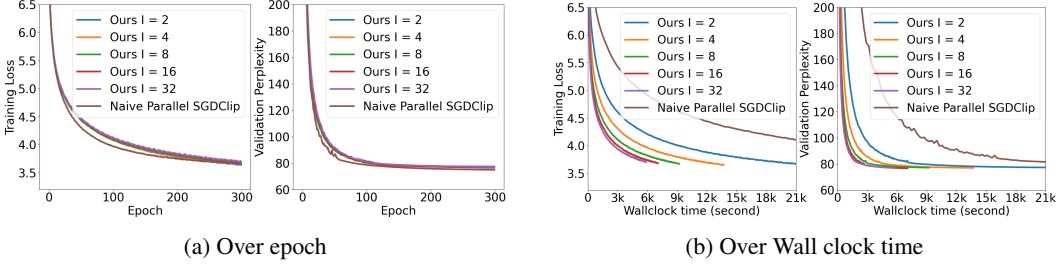

Figure 3: Algorithm 1 with different $I$: Training loss and validation perplexity v.s. (Left) epoch and (right) wall clock time on training an AWD-LSTM to do language modeling on Wikitext-2.

other nodes periodically with node synchronization interval length $I$. To study how communication skipping affects the convergence of Algorithm 1, we run it with $I \in \{2, 4, 8, 16, 32\}$.

**CIFAR-10 classification with ResNet-56.** We train the standard 56-layer ResNet (He et al., 2016) architecture on CIFAR-10. We use SGD with clipping as the baseline algorithm with a stagewise decaying learning rate schedule, following the widely adopted fashion on training the ResNet architecture. Specifically, we use the initial learning rate $\eta = 0.3$, the clipping threshold $\gamma = 1.0$, and decrease the learning rate by a factor of 10 at epoch 80 and 120. The local batch size at each GPU is 64. These parameter settings follow that of Yu et al. (2019a).

The results are illustrated in Figure 1. Figure 1a shows the convergence of training loss and test accuracy v.s. the number of epochs that are jointly accessed by all GPUs. This means that, if the x-axis value is 8, then each GPU runs 1 epoch of training data. The same convention applied to all other figures for multiple GPU training in this paper. Figure 1b verifies our algorithm's advantage of skipping communication by plotting the convergence of training loss and test accuracy v.s. the wall clock time. Overall, we can clearly see that our algorithm matches the baseline epoch-wise but greatly speeds up wall-clock-wise.

**Language modeling with LSTM on Penn Treebank.** We adopt the 3-layer AWD-LSTM (Merity et al., 2018) to do language modeling on Penn Treebank (PTB) dataset (Marcus et al., 1993)(word

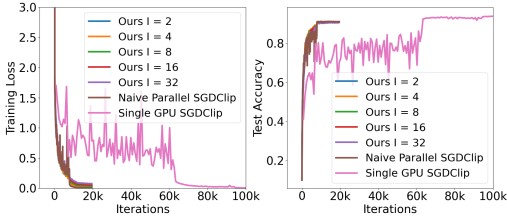 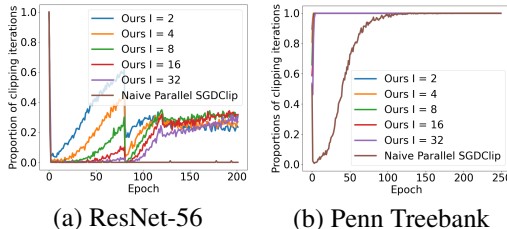

(a) ResNet-56     (b) Penn Treebank

Figure 4: Performance v.s. # of iterations each GPU runs on training ResNet-56 on CIFAR-10 showing the parallel speedup.

Figure 5: Proportions of iterations in each epoch in which clipping is triggered v.s. epochs showing clipping is very frequent.

level). We use SGD with clipping as the baseline algorithm with the initial learning rate $\eta = 30$ and the clipping threshold $\gamma = 7.5$. The local batch size at each GPU is 3. These parameter settings follow that of Zhang et al. (2020).

We report the results in Figure 2. Though we slightly fall behind the baseline epoch-wise in terms of validation perplexity, we do better in training, and gains substantial speedup (2x faster for $I = 16$) wall-clock-wise.

**Language modeling with LSTM on Wikitext-2.** We again adopt the 3-layer AWD-LSTM (Merity et al., 2018) to do language modeling on Wikitext-2 dataset (Marcus et al., 1993)(word level). We use SGD with clipping as the baseline algorithm with the initial learning rate $\eta = 30$ and the clipping threshold $\gamma = 7.5$. The local batch size at each GPU is 10. These parameter settings follow that of Merity et al. (2018).

We report the results in Figure 3. We can match the baseline in both training loss and validation perplexity epoch wise, but we again obtain large speedup (2.5x faster for $I = 16$) wall-clock-wise. This, together with the above two experiments, clearly show our algorithm's effectiveness in speeding up the training in distributed settings. Another observation is that Algorithm 1 can allow relatively large $I$ without hurting the convergence behavior.

### 5.2 VERIFYING PARALLEL SPEEDUP

Figure 4 shows the training loss and test accuracy v.s. the number of iterations. In the distributed setting, one iteration means running one step of Algorithm 1 on all machine; while in the single machine setting, one iteration means running one step of SGD with clipping. In our experiment, we use minibatch size 64 on every GPU in distributed setting to run Algorithm 1, while we also use 64 minibatch size on the single GPU to run SGD with clipping. In Figure 4, we can clearly find that even with $I > 1$, our algorithm still enjoys parallel speedup, since our algorithm requires less number of iterations to converge to the same targets (e.g., training loss, test accuracy). This observation is consistent with our iteration complexity results in Theorem 1.

### 5.3 CLIPPING OPERATION HAPPENS FREQUENTLY

Figure 5 reports the proportion of iterations in each epoch that clipping is triggered. We observe that for our algorithm, clipping happens more frequently than the baseline, especially for NLP tasks. We conjecture that this is because we only used local gradients in each GPU to do the clipping without averaging them across all machines like the baseline did. This leads to more stochasticity of the norm of the gradient in our algorithm than the baseline, and thus causes more clippings to happen. This observation highlights the importance of studying clipping algorithms in the distributed setting. Another interesting observation is that clipping happens much more frequently when training language models than image classification models. Hence this algorithm is presumably more effective in training deep models in NLP tasks.

### 6 CONCLUSION

In this paper, we design a communication-efficient distributed stochastic local gradient clipping algorithm to train deep neural networks. By exploring the relaxed smoothness condition which was shown to be satisfied for certain neural networks, we theoretically prove both the linear speedup property and the improved communication complexity. Our empirical studies show that our algorithm indeed enjoys parallel speedup and greatly improves the runtime performance due to skipping communication rounds in various scenarios.

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

APPENDIX

## A  PROPERTIES OF $(L_0, L_1)$ FUNCTIONS AND USEFUL LEMMAS

**Lemma 4** (Descent inequality in Zhang et al. (2020)). *Let $f$ be $(L_0, L_1)$-smooth, and $c > 0$ be a constant. For any $\mathbf{x}_k$ and $\mathbf{x}_{k+1}$, as long as $\|\mathbf{x}_k - \mathbf{x}_{k+1}\| \leq c/L_1$, we have*

$$f(\mathbf{x}_{k+1}) \leq f(\mathbf{x}_k) + \langle \nabla f(\mathbf{x}_k), \mathbf{x}_{k+1} - x_k \rangle + \frac{AL_0 + BL_1\|\nabla f(\mathbf{x}_k)\|}{2}\|\mathbf{x}_{k+1} - \mathbf{x}_k\|^2, \quad (4)$$

*where $A = 1 + e^c - \frac{e^c - 1}{c}$ and $B = \frac{e^c - 1}{c}$.*

**Lemma 5** (Zhang et al. (2020)). *Let $f$ be $(L_0, L_1)$-smooth, and $c > 0$ be a constant. For any $\mathbf{x}_k$ and $\mathbf{x}_{k+1}$, as long as $\|\mathbf{x}_k - \mathbf{x}_{k+1}\| \leq c/L_1$, we have*

$$\|\nabla f(\mathbf{x}_{k+1}) - \nabla f(\mathbf{x}_k)\| \leq (AL_0 + BL_1\|\nabla f(\mathbf{x}_k)\|) \|\mathbf{x}_{k+1} - \mathbf{x}_k\|, \quad (5)$$

*where $A = 1 + e^c - \frac{e^c - 1}{c}$ and $B = \frac{e^c - 1}{c}$.*

**Lemma 6** (Zhang et al. (2020)). *Let $\mu \geq 0$ be a real constant. For any vector $u$ and $v$,*

$$-\frac{\langle \mathbf{u}, \mathbf{v} \rangle}{\|\mathbf{v}\|} \leq -\mu\|\mathbf{u}\| - (1 - \mu)\|\mathbf{v}\| + (1 + \mu)\|\mathbf{v} - \mathbf{u}\|. \quad (6)$$

## B  PROOF OF THEOREM 1

During the proof of Theorem 1, we need the following simple fact for our algorithm.

**Fact**   Recall the definition of $\bar{\mathbf{x}}_t = \frac{1}{N}\sum_{i=1}^{N} \mathbf{x}_t^i$ for a fixed iteration $t$, Algorithm 1 immediately gives us $\bar{\mathbf{x}}_{t+1} - \bar{\mathbf{x}}_t = -\frac{1}{N}\sum_{i=1}^{N} \min\left(\eta, \frac{\gamma}{\|\nabla F_i(\mathbf{x}_t^i; \xi_t^i)\|}\right) \nabla F_i(\mathbf{x}_t^i; \xi_t^i)$, and hence $\|\bar{\mathbf{x}}_{t+1} - \bar{\mathbf{x}}_t\|^2 \leq \gamma^2$ holds for any $t$.

### B.1  PROOF OF LEMMA 1

**Lemma 1 restated**   Under Assumption 1, Algorithm 1 ensures

$$\|\bar{\mathbf{x}}_t - \mathbf{x}_t^i\| \leq 2\gamma I, \forall i, \forall t \quad \text{almost surely.} \quad (7)$$

*Proof.* Fix $t \geq 1$ and $i$. Note that Algorithm 1 calculates the node average every $I$ iterations. Let $t_0 \leq t$ to be the largest multiple of $I$. Then we know $\bar{\mathbf{x}}_{t_0} = \mathbf{x}_t^i$ for all $i$. (Note that $t - t_0 \leq I$). We further notice that

$$\mathbf{x}_t^i = \bar{\mathbf{x}}_{t_0} - \sum_{\tau=t_0}^{t-1} \min\left(\eta, \frac{\gamma}{\|\nabla F_i(\mathbf{x}_t^i, \xi_t^i)\|}\right) \nabla F_i(\mathbf{x}_t^i, \xi_t^i).$$

By the definition of $\bar{\mathbf{x}}_t$, we have

$$\bar{\mathbf{x}}_t = \bar{\mathbf{x}}_{t_0} - \sum_{\tau=t_0}^{t-1} \frac{1}{N} \sum_{j=1}^{N} \min\left(\eta, \frac{\gamma}{\|\nabla F_j(\mathbf{x}_t^j, \xi_t^j)\|}\right) \nabla F_j(\mathbf{x}_t^j, \xi_t^j)$$

Thus, we have

$$\|\bar{\mathbf{x}}_t - \mathbf{x}_t^i\|^2$$

$$= \left\| \sum_{\tau=t_0}^{t-1} \min\left(\eta, \frac{\gamma}{\|\nabla F_i(\mathbf{x}_t^i, \xi_t^i)\|}\right) \nabla F_i(\mathbf{x}_t^i, \xi_t^i) - \sum_{\tau=t_0}^{t-1} \frac{1}{N} \sum_{j=1}^{N} \min\left(\eta, \frac{\gamma}{\|\nabla F_j(\mathbf{x}_t^j, \xi_t^j)\|}\right) \nabla F_i(\mathbf{x}_t^j, \xi_t^j) \right\|^2$$

$$\leq 2 \left\| \sum_{\tau=t_0}^{t-1} \min\left(\eta, \frac{\gamma}{\|\nabla F_i(\mathbf{x}_t^i, \xi_t^i)\|}\right) \nabla F_i(\mathbf{x}_t^i, \xi_t^i) \right\|^2 + 2 \left\| \sum_{\tau=t_0}^{t-1} \frac{1}{N} \sum_{j=1}^{N} \min\left(\eta, \frac{\gamma}{\|\nabla F_j(\mathbf{x}_t^j, \xi_t^j)\|}\right) \nabla F_j(\mathbf{x}_t^j, \xi_t^j) \right\|^2$$

$$\leq 2(t-t_0) \sum_{\tau=t_0}^{t-1} \left\| \min\left(\eta, \frac{\gamma}{\|\nabla F_i(\mathbf{x}_t^i, \xi_t^i)\|}\right) \nabla F_i(\mathbf{x}_t^i, \xi_t^i) \right\|^2 + \frac{2(t-t_0)}{N^2} \sum_{\tau=t_0}^{t-1} \left\| \sum_{j=1}^{N} \min\left(\eta, \frac{\gamma}{\|\nabla F_j(\mathbf{x}_t^j, \xi_t^j)\|}\right) \nabla F_j(\mathbf{x}_t^j, \xi_t^j) \right\|^2$$

$$\leq 2\gamma^2(t-t_0) \sum_{\tau=t_0}^{t-1} \left\| \frac{\nabla F_i(\mathbf{x}_t^i, \xi_t^i)}{\|\nabla F_i(\mathbf{x}_t^i, \xi_t^i)\|} \right\|^2 + \frac{2(t-t_0)}{N^2} N \sum_{\tau=t_0}^{t-1} \sum_{j=1}^{N} \gamma^2 \left\| \frac{\nabla F_j(\mathbf{x}_t^j, \xi_t^j)}{\|\nabla F_i(\mathbf{x}_t^j, \xi_t^j)\|} \right\|^2$$

$$\leq 2\gamma^2 I^2 + \frac{2(t-t_0)}{N} \sum_{\tau=t_0+1}^{t} \sum_{j=1}^{N} \gamma^2$$

$$\leq 4\gamma^2 I^2. \tag{8}$$

The technique we mainly use in this proof is the inequality $\|\sum_{j=1}^{K} \mathbf{z}_j\|^2 \leq K \sum_{j=1}^{K} \|\mathbf{z}_j\|^2$ for any $K$ vectors $\mathbf{z}_1, \ldots, \mathbf{z}_K$. Taking the square root for both sides, then we obtain the desired result. $\qquad\square$

### B.2 PROOF OF LEMMA 2

Note that $J(t) = \{i \in [1, \ldots, N] \mid \|\nabla F_i(\mathbf{x}_t^i, \xi_t^i)\| \geq \frac{\gamma}{\eta}\}$.

**Lemma 2 restated** If $AL_0\eta \leq 1/2$ and $2\gamma I \leq c/L_1$ for some $c > 0$, then we have

$$\mathbb{E}[f(\bar{\mathbf{x}}_{t+1}) - f(\bar{\mathbf{x}}_t)]$$
$$\leq \frac{1}{N} \mathbb{E} \sum_{i \in J(t)} \left[ -\frac{2\gamma}{5} \|\nabla f(\bar{\mathbf{x}}_t)\| - \frac{3\gamma^2}{5\eta} + \frac{7\gamma}{5} \|\nabla F_i(\mathbf{x}_t^i; \xi_t^i) - \nabla f(\bar{\mathbf{x}}_t)\| + AL_0\gamma^2 + \frac{BL_1\gamma^2 \|\nabla f(\bar{\mathbf{x}}_t)\|}{2} \right]$$
$$+ \frac{1}{N} \sum_{i \in \bar{J}(t)} \mathbb{E} \left[ -\frac{\eta}{2} \|\nabla f(\bar{\mathbf{x}}_t)\|^2 + 4\gamma^2 I^2 A^2 L_0^2 \eta + 4\gamma^2 I^2 B^2 L_1^2 \eta \|\nabla f(\bar{\mathbf{x}}_t)\|^2 + \frac{AL_0\eta^2\sigma^2}{N} + \frac{BL_1\gamma^2 \|\nabla f(\bar{\mathbf{x}}_t)\|}{2} \right],$$

where $A = 1 + e^c - \frac{e^c - 1}{c}$ and $B = \frac{e^c - 1}{c}$.

*Proof.* By the $(L_0, L_1)$-smoothness and invoking Lemma 4 and noting that $\|\bar{\mathbf{x}}_{t+1} - \bar{\mathbf{x}}_t\| \leq \gamma$ for any $t$, we have

$$\mathbb{E}[f(\bar{\mathbf{x}}_{t+1}) - f(\bar{\mathbf{x}}_t)]$$
$$\leq \mathbb{E} \langle \nabla f(\bar{\mathbf{x}}_t), \bar{\mathbf{x}}_{t+1} - \bar{\mathbf{x}}_t \rangle + \frac{(AL_0 + BL_1 \|\nabla f(\bar{\mathbf{x}}_t)\|)}{2} \mathbb{E} \|\bar{\mathbf{x}}_{t+1} - \bar{\mathbf{x}}_t\|^2$$
$$\leq -\gamma \mathbb{E} \left\langle \frac{1}{N} \sum_{i \in J(t)} \frac{\nabla F_i(\mathbf{x}_t^i, \xi_t^i)}{\|\nabla F_i(\mathbf{x}_t^i; \xi_t^i)\|}, \nabla f(\bar{\mathbf{x}}_t) \right\rangle - \eta \mathbb{E} \left\langle \frac{1}{N} \sum_{i \in \bar{J}(t)} \nabla F_i(\mathbf{x}_t^i, \xi_t^i), \nabla f(\bar{\mathbf{x}}_t) \right\rangle \tag{9}$$
$$+ \frac{AL_0}{2} \mathbb{E} \|\bar{\mathbf{x}}_{t+1} - \bar{\mathbf{x}}_t\|^2 + \frac{BL_1 \|\nabla f(\bar{\mathbf{x}}_t)\| \gamma^2}{2}.$$

For each iteration $t$, we first consider the case where $i \in J(t)$, i.e. $\|\nabla F_i(\mathbf{x}_t^i, \xi_t^i)\| \geq \frac{\gamma}{\eta}$. We use Lemma 6 with $\mu = \frac{2}{5}$ to obtain the following result

$$-\gamma \frac{\langle \nabla F_i(\mathbf{x}_t^i, \xi_t^i), \nabla f(\bar{\mathbf{x}}_t) \rangle}{\|\nabla F_i(\mathbf{x}_t^i; \xi_t^i)\|} \leq -\frac{2\gamma}{5} \|\nabla f(\bar{\mathbf{x}}_t)\| - \frac{3\gamma^2}{5\eta} + \frac{7\gamma}{5} \|\nabla F_i(\mathbf{x}_t^i; \xi_t^i) - \nabla f(\bar{\mathbf{x}}_t)\|. \tag{10}$$

Taking the expectation for both sides of above inequality, then we obtain

$$
-\gamma\mathbb{E}\left\langle \frac{1}{N}\sum_{i\in J(t)}\frac{\nabla F_i(\mathbf{x}_t^i,\xi_t^i)}{\|\nabla F_i(\mathbf{x}_t^i;\xi_t^i)\|},\nabla f(\bar{\mathbf{x}}_t)\right\rangle \le \frac{1}{N}\mathbb{E}\sum_{i\in J(t)}\left[-\frac{2\gamma}{5}\|\nabla f(\bar{\mathbf{x}}_t)\|-\frac{3\gamma^2}{5\eta}+\frac{7\gamma}{5}\|\nabla F_i(\mathbf{x}_t^i;\xi_t^i)-\nabla f(\bar{\mathbf{x}}_t)\|\right].
$$
(11)

Next, we consider the case in which $i\in\bar{J}(t)$, i.e. $\|\nabla F_i(\mathbf{x}_t^i,\xi_t^i)\|\le\frac{\gamma}{\eta}$ at fixed iteration $t$. Denote $\xi^{[t-1]}$ by the $\sigma$-algebra generated by all random variables until $(t-1)$-th iteration, i.e., $\xi^{[t-1]}=\sigma(\xi_0^1,\ldots,\xi_0^N,\xi_1^1,\ldots,\xi_1^N,\ldots,\xi_{t-1}^1,\ldots,\xi_{t-1}^N)$. We notice that

$$
-\eta\mathbb{E}\left\langle\frac{1}{N}\sum_{i\in\bar{J}(t)}\nabla F_i(\mathbf{x}_t^i,\xi_t^i),\frac{\mathbb{E}\left[|\bar{J}(t)|\right]}{N}\nabla f(\bar{\mathbf{x}}_t)\right\rangle = -\eta\mathbb{E}\left[\mathbb{E}\left\langle\frac{1}{N}\sum_{i\in\bar{J}(t)}\nabla F_i(\mathbf{x}_t^i,\xi_t^i),\frac{\mathbb{E}\left[|\bar{J}(t)|\right]}{N}\nabla f(\bar{\mathbf{x}}_t)\,\bigg|\,\xi^{[t-1]}\right\rangle\right]
$$

$$
= -\eta\mathbb{E}\left[\left\langle\mathbb{E}\left[\frac{1}{N}\sum_{i\in\bar{J}(t)}\nabla F_i(\mathbf{x}_t^i,\xi_t^i)\,\bigg|\,\xi^{[t-1]}\right],\frac{\mathbb{E}\left[|\bar{J}(t)|\right]}{N}\nabla f(\bar{\mathbf{x}}_t)\right\rangle\right]
$$

$$
= -\eta\mathbb{E}\left[\left\langle\mathbb{E}\left[\frac{1}{N}\sum_{i=1}^N\nabla F_i(\mathbf{x}_t^i,\xi_t^i)\mathbb{I}(i\in\bar{J}(t))\,\bigg|\,\xi^{[t-1]}\right],\frac{\mathbb{E}\left[|\bar{J}(t)|\right]}{N}\nabla f(\bar{\mathbf{x}}_t)\right\rangle\right]
$$

$$
\overset{(a)}{=} -\eta\mathbb{E}\left[\left\langle\frac{1}{N}\sum_{i=1}^N p_i\nabla f_i(\mathbf{x}_t^i),\frac{\mathbb{E}\left[|\bar{J}(t)|\right]}{N}\nabla f(\bar{\mathbf{x}}_t)\right\rangle\right]
$$

$$
\overset{(b)}{=} -\frac{\eta}{2}\mathbb{E}\left[\left\|\frac{\mathbb{E}\left[|\bar{J}(t)|\right]}{N}\nabla f(\bar{\mathbf{x}}_t)\right\|^2+\left\|\frac{1}{N}\sum_{i=1}^N p_i\nabla f_i(\mathbf{x}_t^i)\right\|^2-\left\|\frac{1}{N}\sum_{i=1}^N p_i\nabla f_i(\mathbf{x}_t^i)-\frac{\mathbb{E}\left[|\bar{J}(t)|\right]}{N}\nabla f(\bar{\mathbf{x}}_t)\right\|^2\right]
$$

$$
= -\frac{\eta(\mathbb{E}|\bar{J}(t)|)^2}{2N^2}\mathbb{E}\|\nabla f(\bar{\mathbf{x}}_t)\|^2-\frac{\eta}{2}\mathbb{E}\left\|\frac{1}{N}\sum_{i=1}^N p_i\nabla f_i(\mathbf{x}_t^i)\right\|^2+\frac{\eta}{2}\mathbb{E}\left\|\frac{1}{N}\sum_{i=1}^N p_i(\nabla f(\bar{\mathbf{x}}_t)-\nabla f_i(\mathbf{x}_t^i))\right\|^2
$$

$$
\overset{(c)}{\le} -\frac{\eta(\mathbb{E}|\bar{J}(t)|)^2}{2N^2}\mathbb{E}\|\nabla f(\bar{\mathbf{x}}_t)\|^2-\frac{\eta}{2}\mathbb{E}\left\|\frac{1}{N}\sum_{i=1}^N p_i\nabla f_i(\mathbf{x}_t^i)\right\|^2+\frac{\eta}{2}\frac{(\mathbb{E}|\bar{J}(t)|)^2}{N^2}\frac{1}{N}\mathbb{E}\sum_{i=1}^N\|(\nabla f_i(\bar{\mathbf{x}}_t)-\nabla f_i(\mathbf{x}_t^i))\|^2
$$

$$
\overset{(d)}{\le} -\frac{\eta(\mathbb{E}|\bar{J}(t)|)^2}{2N^2}\mathbb{E}\|\nabla f(\bar{\mathbf{x}}_t)\|^2-\frac{\eta}{2}\mathbb{E}\left\|\frac{1}{N}\sum_{i=1}^N p_i\nabla f_i(\mathbf{x}_t^i)\right\|^2+\frac{\eta}{2}\frac{(\mathbb{E}|\bar{J}(t)|)^2}{N^2}\mathbb{E}\left[4\gamma^2 I^2\left(AL_0+BL_1\|\nabla f(\bar{\mathbf{x}}_t)\|\right)^2\right]
$$

$$
\overset{(e)}{\le} -\frac{\eta(\mathbb{E}|\bar{J}(t)|)^2}{2N^2}\mathbb{E}\|\nabla f(\bar{\mathbf{x}}_t)\|^2-\frac{\eta}{2}\mathbb{E}\left\|\frac{1}{N}\sum_{i=1}^N p_i\nabla f_i(\mathbf{x}_t^i)\right\|^2+\frac{\eta}{2}\frac{(\mathbb{E}|\bar{J}(t)|)^2}{N^2}\mathbb{E}\left[2\gamma^2 I^2\left(2A^2 L_0^2+2B^2 L_1^2\|\nabla f(\bar{\mathbf{x}}_t)\|^2\right)\right]
$$

$$
\le -\frac{\eta(\mathbb{E}|\bar{J}(t)|)^2}{2N^2}\mathbb{E}\|\nabla f(\bar{\mathbf{x}}_t)\|^2-\frac{\eta}{2}\mathbb{E}\left\|\frac{1}{N}\sum_{i=1}^N p_i\nabla f_i(\mathbf{x}_t^i)\right\|^2+\frac{(\mathbb{E}|\bar{J}(t)|)^2}{N^2}\mathbb{E}\left[4\gamma^2 I^2 A^2 L_0^2\eta+4\gamma^2 I^2 B^2 L_1^2\eta\|\nabla f(\bar{\mathbf{x}}_t)\|^2\right],
$$
(12)

where (a) holds due to Lemma 8 with $A_i=\nabla F_i(\mathbf{x}_t^i;\xi_t^i)$, $B_i=\mathbb{I}(i\in\bar{J}(t))$, $0\le p_i\le 1$ is the corresponding constant defined in Lemma 8 and $\sum_{i=1}^N p_i=\mathbb{E}\left[|\bar{J}(t)|\right]$; (b) holds by using the basic identity $\langle\mathbf{x}_1,\mathbf{x}_2\rangle=\frac{1}{2}\left(\|\mathbf{x}_1\|^2+\|\mathbf{x}_2\|^2-\|\mathbf{x}_1-\mathbf{x}_2\|^2\right)$ for any two vectors $\mathbf{x}_1,\mathbf{x}_2$; (c) holds due to Jensen's inequality and $\sum_{i=1}^N p_i=\mathbb{E}|\bar{J}(t)|$. (d) holds due to Lemma 1 and Lemma 5, (e) holds since $(a+b)^2\le 2a^2+2b^2$.

Next, we compute $\mathbb{E}\|\bar{\mathbf{x}}_{t+1} - \bar{\mathbf{x}}_t\|^2$. By definition, we have

$$\mathbb{E}\|\bar{\mathbf{x}}_{t+1} - \bar{\mathbf{x}}_t\|^2 = \mathbb{E}\left\|\frac{1}{N}\sum_{i=1}^{N}\min\left(\eta, \frac{\gamma}{\|\nabla F_i(\mathbf{x}_t^i;\xi_t^i)\|}\right)\nabla F_i(\mathbf{x}_t^i,\xi_t^i)\right\|^2$$

$$= \frac{1}{N^2}\mathbb{E}\left\|\sum_{i=1}^{N}\min\left(\eta, \frac{\gamma}{\|\nabla F_i(\mathbf{x}_t^i;\xi_t^i)\|}\right)\nabla F_i(\mathbf{x}_t^i,\xi_t^i)\right\|^2 = \frac{1}{N^2}\mathbb{E}\left\|\gamma\sum_{i\in J(t)}\frac{\nabla F_i(\mathbf{x}_t^i,\xi_t^i)}{\|\nabla F_i(\mathbf{x}_t^i;\xi_t^i)\|} + \eta\sum_{i\in\bar{J}(t)}\eta\nabla F_i(\mathbf{x}_t^i;\xi_t^i)\right\|^2$$

$$\leq 2\gamma^2\mathbb{E}\left\|\frac{1}{N}\sum_{i\in J(t)}\frac{\nabla F_i(\mathbf{x}_t^i,\xi_t^i)}{\|\nabla F_i(\mathbf{x}_t^i;\xi_t^i)\|}\right\|^2 + 2\eta^2\mathbb{E}\left\|\frac{1}{N}\sum_{i=1}^{N}\nabla F_i(\mathbf{x}_t^i,\xi_t^i)\mathbb{I}\left(i\in\bar{J}(t)\right)\right\|^2$$

$$\leq \frac{2\gamma^2}{N^2}\mathbb{E}\left\|\sum_{i\in J(t)}\frac{\nabla F_i(\mathbf{x}_t^i,\xi_t^i)}{\|\nabla F_i(\mathbf{x}_t^i;\xi_t^i)\|}\right\|^2 + 2\eta^2\mathbb{E}\left\|\frac{1}{N}\sum_{i=1}^{N}\nabla F_i(\mathbf{x}_t^i,\xi_t^i)\mathbb{I}\left(i\in\bar{J}(t)\right) - \mathbb{E}\left[\frac{1}{N}\sum_{i=1}^{N}\nabla F_i(\mathbf{x}_t^i,\xi_t^i)\mathbb{I}\left(i\in\bar{J}(t)\right)\right]\right\|^2$$

$$+ 2\eta^2\left\|\mathbb{E}\left(\frac{1}{N}\sum_{i=1}^{N}\nabla F_i(\mathbf{x}_t^i,\xi_t^i)\mathbb{I}\left(i\in\bar{J}(t)\right)\right)\right\|^2$$

$$\overset{(a)}{\leq} \frac{2\gamma^2(\mathbb{E}|J(t)|)^2}{N^2} + 2\eta^2\mathbb{E}\left\|\frac{1}{N}\sum_{i=1}^{N}\nabla F_i(\mathbf{x}_t^i,\xi_t^i) - \mathbb{E}\left[\frac{1}{N}\sum_{i=1}^{N}\nabla F_i(\mathbf{x}_t^i,\xi_t^i)\right]\right\|^2 + 2\eta^2\left\|\mathbb{E}\left(\frac{1}{N}\sum_{i=1}^{N}\nabla F_i(\mathbf{x}_t^i,\xi_t^i)\mathbb{I}\left(i\in\bar{J}(t)\right)\right)\right\|^2$$

$$\leq \frac{2\gamma^2(\mathbb{E}|J(t)|)^2}{N^2} + \frac{2\eta^2\sigma^2}{N} + 2\eta^2\left\|\mathbb{E}\left(\frac{1}{N}\sum_{i=1}^{N}\nabla F_i(\mathbf{x}_t^i,\xi_t^i)\mathbb{I}\left(i\in\bar{J}(t)\right)\right)\right\|^2$$

$$\overset{(b)}{=} \frac{2\gamma^2(\mathbb{E}|J(t)|)^2}{N^2} + \frac{2\eta^2\sigma^2}{N} + 2\eta^2\left\|\mathbb{E}\left(\sum_{i=1}^{N}p_i\nabla f_i(\mathbf{x}_t^i)\right)\right\|^2$$

(13)

where (a) holds due to Lemma 7 with $A_i = \nabla F_i(\mathbf{x}_t^i;\xi_t^i)$, $B_i = \mathbb{I}\left(i\in\bar{J}(t)\right)$; (b) holds due to Lemma 8 with $A_i = \nabla F_i(\mathbf{x}_t^i;\xi_t^i)$, $B_i = \mathbb{I}(i\in\bar{J}(t))$, $0 \leq p_i \leq 1$ is the corresponding constant defined in Lemma 8 and $\sum_{i=1}^{N}p_i = \mathbb{E}\left[|\bar{J}(t)|\right]$.

Substituting (11), (12) and (13) into (9) yields

$$\mathbb{E}[f(\bar{\mathbf{x}}_{t+1}) - f(\bar{\mathbf{x}}_t)]$$

$$\leq \frac{1}{N}\mathbb{E}\sum_{i\in J(t)}\left[-\frac{2\gamma}{5}\|\nabla f(\bar{\mathbf{x}}_t)\| - \frac{3\gamma^2}{5\eta} + \frac{7\gamma}{5}\|\nabla F_i(\mathbf{x}_t^i;\xi_t^i) - \nabla f(\bar{\mathbf{x}}_t)\|\right] - \frac{\eta\mathbb{E}|\bar{J}(t)|}{2N}\mathbb{E}\|\nabla f(\bar{\mathbf{x}}_t)\|^2$$

$$-\frac{\eta}{2}\frac{N}{\mathbb{E}|\bar{J}(t)|}\mathbb{E}\left\|\frac{1}{N}\sum_{i=1}^{N}p_i\nabla f_i(\mathbf{x}_t^i)\right\|^2 + \frac{\mathbb{E}|\bar{J}(t)|}{N}\mathbb{E}\left[4\gamma^2 I^2 A^2 L_0^2\eta + 4\gamma^2 I^2 B^2 L_1^2\eta\|\nabla f(\bar{\mathbf{x}}_t)\|^2\right]$$

$$+\frac{AL_0\gamma^2(\mathbb{E}|J(t)|)^2}{N^2} + \frac{AL_0\eta^2\sigma^2}{N} + AL_0\eta^2\left\|\mathbb{E}\left(\sum_{i=1}^{N}p_i\nabla f_i(\mathbf{x}_t^i)\right)\right\|^2 + \frac{BL_1\gamma^2\mathbb{E}\|\nabla f(\bar{\mathbf{x}}_t)\|}{2}$$

$$\leq \frac{1}{N}\mathbb{E}\sum_{i\in J(t)}\left[-\frac{2\gamma}{5}\|\nabla f(\bar{\mathbf{x}}_t)\| - \frac{3\gamma^2}{5\eta} + \frac{7\gamma}{5}\|\nabla F_i(\mathbf{x}_t^i;\xi_t^i) - \nabla f(\bar{\mathbf{x}}_t)\| + AL_0\gamma^2 + \frac{BL_1\gamma^2\|\nabla f(\bar{\mathbf{x}}_t)\|}{2} + \frac{AL_0\eta^2\sigma^2}{N}\right]$$

$$+\frac{1}{N}\mathbb{E}\sum_{i\in\bar{J}(t)}\left[-\frac{\eta}{2}\|\nabla f(\bar{\mathbf{x}}_t)\|^2 + 4\gamma^2 I^2 A^2 L_0^2\eta + 4\gamma^2 I^2 B^2 L_1^2\eta\|\nabla f(\bar{\mathbf{x}}_t)\|^2 + \frac{AL_0\eta^2\sigma^2}{N} + \frac{BL_1\gamma^2\|\nabla f(\bar{\mathbf{x}}_t)\|}{2}\right],$$

(14)

where the last inequality holds if $AL_0\eta \leq \frac{1}{2}$ and $\mathbb{E}\|X\|^2 \geq \|\mathbb{E}X\|^2$. $\qquad\square$

## B.3 PROOF OF LEMMA 3

**Lemma 3 restated** Suppose Assumption 1 holds. If $2\gamma I \le c/L_1$ for some $c > 0$, then we obtain

$$\frac{1}{N} \sum_{i=1}^{N} \|\nabla F_i(\mathbf{x}_t^i; \xi_t^i) - \nabla f(\bar{\mathbf{x}}_t)\| \le \sigma + \kappa + 2\gamma I(AL_0 + BL_1\|\nabla f(\bar{\mathbf{x}}_t)\|) \quad \text{almost surely,}$$

where $A = 1 + e^c - \frac{e^c - 1}{c}$ and $B = \frac{e^c - 1}{c}$.

*Proof.* By using triangular inequality and Lemma 5, we obtain

$$\|\nabla F_i(\mathbf{x}_t^i; \xi_t^i) - \nabla f(\bar{\mathbf{x}}_t)\|$$
$$\le \|\nabla F_i(\mathbf{x}_t^i; \xi_t^i) - \nabla f_i(\mathbf{x}_t^i)\| + \|\nabla f_i(\mathbf{x}_t^i) - \nabla f(\mathbf{x}_t^i)\| + \|\nabla f(\mathbf{x}_t^i) - \nabla f(\bar{\mathbf{x}}_t)\|$$
$$\le \sigma + \|\nabla f_i(\mathbf{x}_t^i) - \nabla f(\mathbf{x}_t^i)\| + 2\gamma I(AL_0 + BL_1\|f(\bar{\mathbf{x}}_t)\|).$$

Summing over $i = 1$ to $N$ and by Assumption 1, we have

$$\frac{1}{N} \sum_{i=1}^{N} \|\nabla F_i(\mathbf{x}_t^i; \xi_t^i) - \nabla f(\bar{\mathbf{x}}_t)\| \le \sigma + \kappa + 2\gamma I(AL_0 + BL_1\|f(\bar{\mathbf{x}}_t)\|).$$

$\square$

## B.4 MAIN PROOF OF THEOREM 1

*Proof.* We consider $0 < \epsilon \le \min(\frac{AL_0}{BL_1}, 0.1)$ be a small enough constant and choose $N \le \min\{\frac{1}{\epsilon}, \frac{14AL_0}{5BL_1\epsilon}\}$, $\gamma \le \frac{N\epsilon}{28(\sigma+\kappa)} \min\{\frac{\epsilon}{AL_0}, \frac{1}{BL_1}\}$, ratio $\gamma/\eta = 5(\sigma + \kappa)$ and $I \le \frac{1}{2N\epsilon}$, we first check several important conditions which were used in previous Lemmas and we will use in our later proof.

$$AL_0\eta \le AL_0 \frac{N\epsilon}{5(\sigma+\kappa) \cdot 28(\sigma+\kappa)} \frac{\epsilon}{AL_0} \overset{(a)}{\le} \frac{\epsilon}{140(\sigma+\kappa)^2} \overset{(b)}{\le} \frac{\epsilon}{140} \le \frac{1}{2},$$

where (a) holds because of $N\epsilon \le 1$ and (b) is true due to $\sigma + \kappa \ge 1$. Note that $N\epsilon \le 1$, $\sigma + \kappa \ge 1$, $A \ge 1$, $B \ge 1$, and so we have

$$2\gamma I \le 2 \frac{N\epsilon}{28(\sigma+\kappa)} \frac{1}{BL_1} \frac{1}{2N\epsilon} \le \frac{1}{28L_1} = \frac{c}{L_1},$$

where $c = \frac{1}{28}$. Recall the descent inequality (Lemma 4), we can explicitly define

$$A = 1 + e^{1/28} - \frac{e^{1/28} - 1}{1/28}$$

$$B = \frac{e^{1/28} - 1}{1/28}.$$

We can show that our choice of $\epsilon$ guarantees that $\frac{\epsilon}{AL_0} \le \frac{1}{BL_1}$. Then, the upper bound of the parameter $\gamma$ becomes $\frac{N\epsilon^2}{28AL_0(\sigma+\kappa)}$. Based on Lemma 2 and Lemma 3, we take summation over $t$

and obtain

$$
\mathbb{E}\left[\sum_{t=0}^{T-1} f(\bar{\mathbf{x}}_{t+1}) - f(\bar{\mathbf{x}}_t)\right]
$$

$$
\overset{(a)}{\leq} \frac{1}{N}\sum_{t=0}^{T-1}\mathbb{E}\sum_{i\in J(t)}\left[P_1 + P_2\|\nabla f(\bar{\mathbf{x}}_t)\|\right]
$$

$$
+\frac{1}{N}\sum_{t=0}^{T-1}\sum_{i\in\bar{J}(t)}\mathbb{E}\left[-\frac{\eta}{2}\|\nabla f(\bar{\mathbf{x}}_t)\|^2 + 4\gamma^2 I^2 A^2 L_0^2\eta + 4\gamma^2 I^2 B^2 L_1^2\eta\|\nabla f(\bar{\mathbf{x}}_t)\|^2 + \frac{AL_0\eta^2\sigma^2}{N} + \frac{BL_1\gamma^2\|\nabla f(\bar{\mathbf{x}}_t)\|}{2}\right]
$$

$$
\overset{(b)}{\leq} \frac{1}{N}\sum_{t=0}^{T-1}\mathbb{E}\sum_{i\in J(t)}\left[P_1 + P_2\|\nabla f(\bar{\mathbf{x}}_t)\|\right]
$$

$$
+\frac{1}{N}\sum_{t=0}^{T-1}\sum_{i\in\bar{J}(t)}\mathbb{E}\left[-\frac{\eta}{4}\|\nabla f(\bar{\mathbf{x}}_t)\|^2 + 4\gamma^2 I^2 A^2 L_0^2\eta + \frac{AL_0\eta^2\sigma^2}{N} + \frac{BL_1\gamma^2\|\nabla f(\bar{\mathbf{x}}_t)\|}{2}\right],
$$

$$
\tag{15}
$$

where (a) holds due to Lemma 2, and

$$
P_1 = -\frac{3\gamma^2}{5\eta} + \frac{7\gamma}{5}(\sigma+\kappa) + \frac{14\gamma^2 IAL_0}{5} + AL_0\gamma^2 + \frac{AL_0\eta^2\sigma^2}{N},
$$

$$
P_2 = -\frac{2\gamma}{5} + \frac{14\gamma^2 IBL_1}{5} + \frac{BL_1\gamma^2}{2},
$$

(b) is derived by computing

$$
4\gamma^2 I^2 B^2 L_1^2\eta \leq \frac{4N^2\epsilon^4 I^2 B^2 L_1^2\eta}{28^2(\sigma+\kappa)^2 A^2 L_0^2} \leq \frac{4B^2 L_1^2\epsilon^4\eta}{28^2(\sigma+\kappa)^2 A^2 L_0^2}\underbrace{\frac{28^2 A^2 L_0^2}{25 B^2 L_1^2\epsilon^2}}_{\text{bound of } N^2}\underbrace{\frac{1}{4N^2\epsilon^2}}_{\text{bound of } I^2} \leq \frac{\eta}{4} \tag{16}
$$

with $N \leq \min\{\frac{1}{\epsilon}, \frac{14AL_0}{5BL_1\epsilon}\}$, $\gamma \leq \frac{N\epsilon}{28(\sigma+\kappa)}\min\{\frac{\epsilon}{AL_0}, \frac{1}{BL_1}\}$ and $I \leq \frac{1}{2N\epsilon}$.

Now we give the upper bound of $P_1$ and $P_2$.

From the choice of $N \leq \min\{\frac{1}{\epsilon}, \frac{14AL_0}{5BL_1\epsilon}\}$, $\gamma \leq \frac{N\epsilon^2}{28AL_0(\sigma+\kappa)}$, $I \leq \frac{1}{2N\epsilon}$ and the fixed ratio $\frac{\gamma}{\eta} = 5(\sigma+\kappa)$, and noting that $\sigma+\kappa \geq 1$, we have

$$
P_1 \leq \left(-\frac{3}{5}\cdot 5(\sigma+\kappa) + \frac{7}{5}(\sigma+\kappa) + \frac{\epsilon}{20} + \frac{\epsilon}{28}\right)\gamma \leq -\frac{3}{10}\gamma(\sigma+\kappa) + \frac{AL_0\eta^2\sigma^2}{N}
$$

$$
P_2 \leq \left(-\frac{2}{5} + \frac{1}{20} + \frac{\epsilon}{10}\right)\gamma \leq -\frac{1}{10}\gamma \tag{17}
$$

Plugging (17) back to (15), we obtain

$$
-\Delta \leq -(f(\bar{\mathbf{x}}_0) - f_*) \leq \mathbb{E}f(\bar{\mathbf{x}}_T) - f(\bar{\mathbf{x}}_0) \leq \mathbb{E}\left[\frac{1}{T}\sum_{t=0}^{T-1}\sum_{i\in J(t)} U(\bar{\mathbf{x}}_t) + \frac{1}{T}\sum_{t=0}^{T-1}\sum_{i\in\bar{J}(t)} V(\bar{\mathbf{x}}_t)\right],
$$

where

$$
U(\bar{\mathbf{x}}_t) = -\frac{1}{10}\gamma\|\nabla f(\bar{\mathbf{x}}_t)\| - \frac{3}{10}\gamma(\sigma+\kappa) + \frac{AL_0\eta^2\sigma^2}{N}
$$

$$
V(\bar{\mathbf{x}}_t) = -\frac{\eta}{4}\|\nabla f(\bar{\mathbf{x}}_t)\|^2 + 4\gamma^2 I^2 A^2 L_0^2\eta + \frac{AL_0\eta^2\sigma^2}{N} + \frac{BL_1\gamma^2\|\nabla f(\bar{\mathbf{x}}_t)\|}{2}
$$

$$
\overset{(a)}{\leq} -\frac{\eta\epsilon}{2}\|\nabla f(\bar{\mathbf{x}}_t)\| + \frac{BL_1\gamma^2\|\nabla f(\bar{\mathbf{x}}_t)\|}{2} + \frac{\eta\epsilon^2}{2} + 4\gamma^2 I^2 A^2 L_0^2\eta + AL_0\frac{\eta^2}{N}\sigma^2
$$

$$
\overset{(b)}{\leq} -\frac{\eta\epsilon}{4}\|\nabla f(\bar{\mathbf{x}}_t)\| + \frac{\eta\epsilon^2}{4} + 4\gamma^2 I^2 A^2 L_0^2\eta + AL_0\frac{\eta^2}{N}\sigma^2.
$$

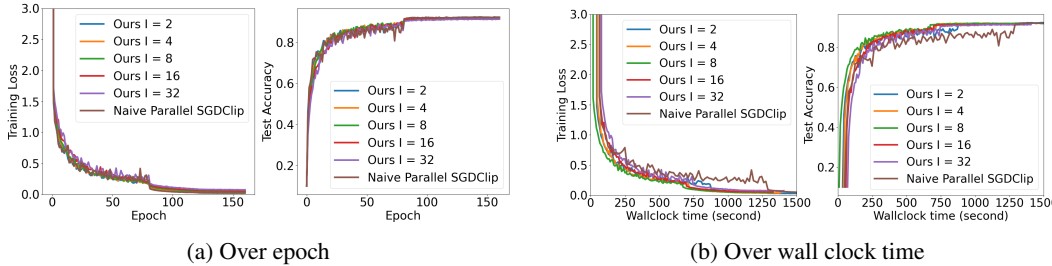

Figure 6: Training loss and test accuracy v.s. (Left) epoch and (right) wall clock time on training a 32 layer Resnet to do image classification on CIFAR10.

Inequality (a) follows by using the standard inequality $x^2 \geq 2\epsilon x - \epsilon^2$ and (b) is true because

$$-\frac{\eta\epsilon}{2} + \frac{BL_1\gamma^2}{2} \leq \frac{-\eta\epsilon}{4}. \tag{18}$$

To check (18), we compute

$$BL_1\gamma^2 \leq BL_1 5\eta(\sigma + \kappa)\frac{N\epsilon^2}{28AL_0(\sigma + \kappa)} = \eta\frac{5BL_1N\epsilon^2}{28AL_0} \leq \frac{\eta\epsilon}{2}, \tag{19}$$

where the last inequality holds because we assume $N \leq \frac{14AL_0}{5BL_1\epsilon}$.

It's clear that $U(\mathbf{x}) \leq V(\mathbf{x})$. Then we obtain

$$-\Delta \leq -(\mathbb{E}f(\bar{\mathbf{x}}_0) - f_*) \leq \mathbb{E}\left[\sum_{t=0}^{T-1} V(\bar{\mathbf{x}}_t)\right] \leq \sum_{t=0}^{T-1}\left[-\frac{\eta\epsilon}{4}\mathbb{E}\|\nabla f(\bar{\mathbf{x}}_t)\| + \frac{\eta\epsilon^2}{4} + 4\gamma^2 I^2 A^2 L_0^2\eta + AL_0\frac{\eta^2}{N}\sigma^2\right].$$

Then we rearrange above inequality and get

$$\frac{1}{T}\sum_{t=1}^{T}\mathbb{E}\|\nabla f(\bar{\mathbf{x}}_t)\| \leq \frac{4\Delta}{\epsilon\eta T} + \epsilon + \frac{4AL_0\eta\sigma^2}{N\epsilon} + \frac{16\gamma^2 I^2 A^2 L_0^2}{\epsilon} < 4\epsilon,$$

as long as $T \geq \frac{560AL_0\Delta(\sigma+\kappa)^2}{N\epsilon^4}$ and $I \leq \frac{1}{2N\epsilon}$. $\qquad\square$

## C    EXPERIMENTS DETAILS AND OTHER RESULTS

### C.1    CIFAR-10 CLASSIFICATION WITH RESNET-32

Apart from the 56 layer Resnet used in Section 5, we also trained the 32 layer Resnet on CIFAR-10. Here, we used a smaller minibatch size of 16 per GPU. We also used SGD with clipping for the baseline algorithm with a stagewise decaying learning rate schedule, We set the initial learning rate $\eta = 0.1$ and the clipping parameter $\gamma = 1.0$. We decrease both $\eta$ and $\gamma$ by a factor of 10 at epoch 80 and 120. These parameter settings follow that of Zhang et al. (2020).

Results reported in Figure 6 again shows that our algorithm can not only match the baseline in both training loss and test accuracy epoch-wise, but is way better in terms of wall clock time for moderate values of $I$.

## D    USEFUL LEMMAS

**Lemma 7.** *Suppose $\{A_i\}_{i=1}^N$ is an sequence of vector-valued independent random variables, $\{B_i\}_{i=1}^N$ is a sequence of independent random variables where $B_i \in \{0, 1\}$ for every $i$. Suppose $\mathbb{E}\|A_i\|^2 < +\infty$ for every $i$. Then we have*

$$\mathbb{E}\left\|\frac{1}{N}\sum_{i=1}^N A_iB_i - \mathbb{E}\left[\frac{1}{N}\sum_{i=1}^N A_iB_i\right]\right\|^2 \leq \mathbb{E}\left\|\frac{1}{N}\sum_{i=1}^N A_i - \mathbb{E}\left[\frac{1}{N}\sum_{i=1}^N A_i\right]\right\|^2.$$

*Proof.* It suffices to show that $\mathbb{E}\|A_i B_i - \mathbb{E}[A_i B_i]\|^2 \leq \mathbb{E}\|A_i - \mathbb{E}[A_i]\|^2$, since both $\{A_i B_i - \mathbb{E}[A_i B_i]\}_{i=1}^N$ and $\{A_i - \mathbb{E}[A_i]\}_{i=1}^N$ are sequences of independent random vectors with zero mean. Define $C_i = 1 - B_i$, and hence $B_i C_i = 0$ with probablity 1. Without loss of generality, we can assume $\mathbb{E}[A_i] = 0$. Note that

$$\mathbb{E}\|A_i - \mathbb{E}[A_i]\|^2 = \mathbb{E}\|A_i B_i + A_i C_i\|^2 = \mathbb{E}\|A_i B_i\|^2 + \mathbb{E}\|A_i C_i\|^2 + 2\mathbb{E}\langle A_i B_i, A_i C_i \rangle$$
$$= \mathbb{E}\|A_i B_i\|^2 + \mathbb{E}\|A_i C_i\|^2 \geq \mathbb{E}\|A_i B_i\|^2 = \mathbb{E}\|A_i B_i - \mathbb{E}[A_i B_i]\|^2 + \|\mathbb{E}(A_i B_i)\|^2$$
$$\geq \mathbb{E}\|A_i B_i - \mathbb{E}[A_i B_i]\|^2.$$

It completes the proof. $\qquad\square$

**Lemma 8.** *Suppose $\{A_i\}_{i=1}^N$ is an sequence of vector-valued independent random variables, $\{B_i\}_{i=1}^N$ is a sequence of independent random variables where $B_i \in \{0, 1\}$ for every $i$. $A_i$ and $B_j$ are independent of each other if $i \neq j$. Then there exists $(p_1, \ldots, p_N)$ where $0 \leq p_i \leq 1$ and $\sum_{i=1}^N p_i = \mathbb{E}\left[\sum_{i=1}^N B_i\right]$ such that*

$$\mathbb{E}\left[\sum_{i=1}^N A_i B_i\right] = \sum_{i=1}^N p_i \mathbb{E}A_i. \tag{20}$$

*Proof.* Without loss of generality, we can assume the sequence $\{B_i\}_{i=1}^N$ is sorted in a decreasing order, i.e., $B_1 \geq B_2 \geq \ldots \geq B_N$, since otherwise we can switch the order of summation in both LHS and RHS of the Equation (20).
Define $\widetilde{T} = \sup\{1 \leq i \leq N : B_i = 1\}$, and define $\mathcal{F}_s = \sigma(A_1, B_1, \ldots, A_s, B_s)$, where $\sigma(\cdot)$ denotes $\sigma$-algebra generated by random variables in the argument. Since $\{\widetilde{T} = s\} \in \mathcal{F}_s$ for all $s$, we know that $\widetilde{T}$ is a stopping time, and $\widetilde{T} \leq N$ almost surely. Hence we have

$$\mathbb{E}\left[\sum_{i=1}^N A_i B_i\right] = \mathbb{E}\left[\sum_{i=1}^{\widetilde{T}} A_i\right] = \mathbb{E}\left[\sum_{i=1}^{+\infty} A_i \mathbb{I}(\widetilde{T} \geq i)\right] = \sum_{i=1}^{+\infty} \mathbb{E}\left[\mathbb{E}\left[A_i \mathbb{I}(\widetilde{T} \geq i)|\mathcal{F}_{t-1}\right]\right].$$

Note that $\mathbb{I}(\widetilde{T} \geq i) = 1 - \mathbb{I}(\widetilde{T} \leq i-1) \in \mathcal{F}_{i-1}$, and $A_i$ is independent of $\mathcal{F}_{i-1}$, we know that

$$\sum_{i=1}^{+\infty} \mathbb{E}\left[\mathbb{E}\left[A_i \mathbb{I}(\widetilde{T} \geq i)|\mathcal{F}_{t-1}\right]\right] = \sum_{i=1}^{+\infty} \mathbb{E}\left[\mathbb{I}(\widetilde{T} \geq i)\mathbb{E}[A_i]\right] = \sum_{i=1}^{+\infty} \mathbb{E}[A_i]\Pr(\widetilde{T} \geq i) = \sum_{i=1}^{N} \mathbb{E}[A_i]\Pr(\widetilde{T} \geq i).$$

Define $p_i = \Pr(\widetilde{T} \geq i)$, we have $0 \leq p_i \leq 1$ and $\sum_{i=1}^N p_i = \mathbb{E}(\widetilde{T}) = \mathbb{E}\left[\sum_{i=1}^N B_i\right]$. Hence (20) is proved. $\qquad\square$

# E   MORE EXPERIMENT RESULTS

## E.1   PARALLEL SPEEDUP ON NLP TASKS

In Section 5.2 we have shown the parallel speedup effects of our algorithm on training a Resnet model on the CIFAR10 dataset. Here we present the same phenomena but on training AWD-LSTMs on doing language modeling on Penn Treebank and Wikitext-2 in Figure 7.

Again, note that in the distributed setting, one iteration means running one step of Algorithm 1 on all machine; while in the single machine setting, one iteration means running one step of SGD with clipping. For Penn Treebank, we use minibatch size 3 on every GPU in distributed setting to run Algorithm 1, while we also use 3 minibatch size on the single GPU to run SGD with clipping. For Wikitext-2, the corresponding minibatch size is 10.

Figure 7 again clearly shows that even with $I > 1$, our algorithm still enjoys parallel speedup, since our algorithm requires less number of iterations to converge to the same targets (e.g., training loss, validation perplexity). This observation is consistent with our iteration complexity results in Theorem 1.

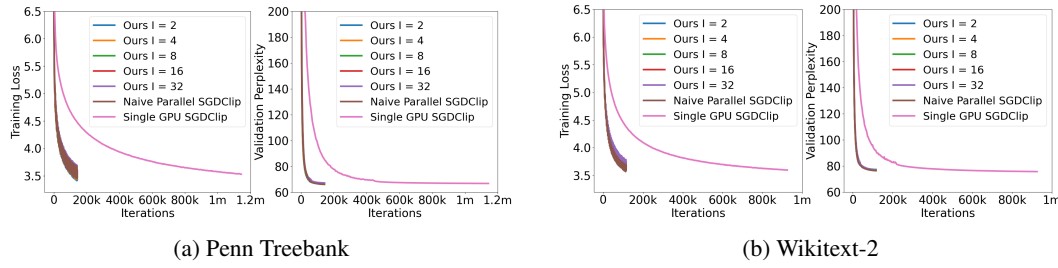

(a) Penn Treebank          (b) Wikitext-2

Figure 7: Performance v.s. # of iterations each GPU runs on training AWD-LSTMs to do language modeling on Penn Treebank (left) and Wikitext-2 (right) showing the parallel speedup.

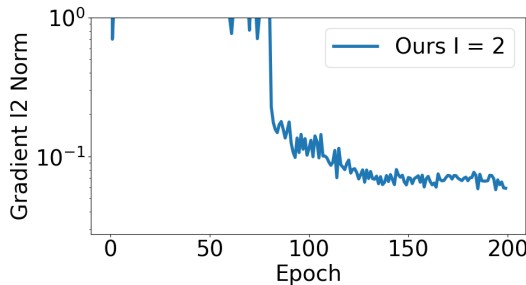

Figure 8: $\ell_2$ norm of gradients over epoch on training ResNet-56 on CIFAR-10.

## E.2 GRADIENT NORM DYNAMICS

The careful reader might notice that in Theorem 1 we require $N \leq O\left(\frac{1}{\epsilon}\right)$ where $\epsilon$ is the desired $\ell_2$ norm of the gradient when converging and wonder if this is too strict and impractical.

Therefore, we draw the dynamics of the $\ell_2$ norm of gradients over epochs for training the Resnet-56 model to do image classification on CIFAR10 in Figure 8. It can be clearly seen that the gradient $\ell_2$ norm decreases to less than $0.1$ after epoch 100 and eventually reaches around $0.06$ at the epoch 200. Since $1/0.06 \approx 17$, this means that the $N = 8$ setting we used in our experiments is consistent with our theory.

## E.3 HETEROGENEOUS SETTING

In the previous experiments, all machines can access the whole dataset. Yet, in practice, each machine might only access a subset of the whole dataset and the data might not be evenly distributed across machines. Therefore, we conduct an experiment on this heterogeneous setting. Specifically, we take the Penn Treebank training set which is a text file and divide it into $8$ non-overlapping consecutive parts which contain $\{10.64\%, 11.17\%, 11.70\%, 12.23\%, 12.77\%, 13.30\%, 13.83\%, 14.36\%\}$ of the whole text file respectively. We then train the AWD-LSTM of the same structure as in Section 5.1 to do language modeling. Considering that the heterogeneous setting introduces big difference compared with the setting we adopted in Section 5.1, we finely tuned the initial learning rate for both the baseline and our algorithm and picked the one that gives the smallest training loss.

The results are reported in Figure 9. Note that, as the size of training data in each machine is different, we draw the training and testing performance curves over the number of iterations each machine runs instead of epochs. This means that, if the x-axis value is $1000$, then each GPU runs $1000$ iterations. We can see that, though the unbalancing and heterogeneity of the training data deteriorates the performance for both the baseline and our algorithm as compared with Figure 2a, our algorithm is still able to surpass the baseline in training losses though it falls slightly behind in testing; meanwhile, our algorithm obtains significant speedup in terms of wall-clock time. This suggests that our algorithm is robust to the heterogeneous setting.

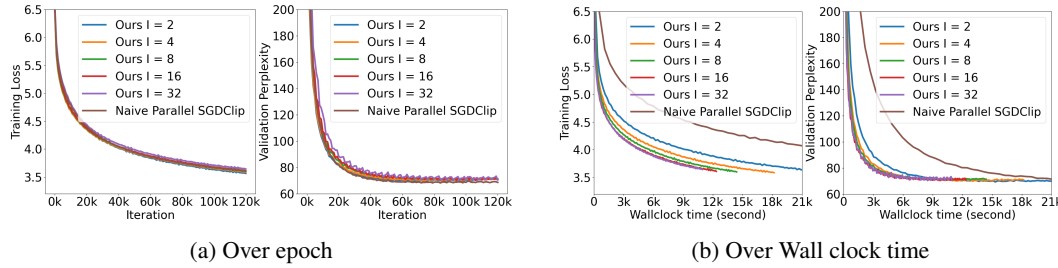

(a) Over epoch              (b) Over Wall clock time

Figure 9: Training an AWD-LSTM to do language modeling on Penn Treebank in a heterogeneous setting where each node accesses only a different subset of the whole dataset.

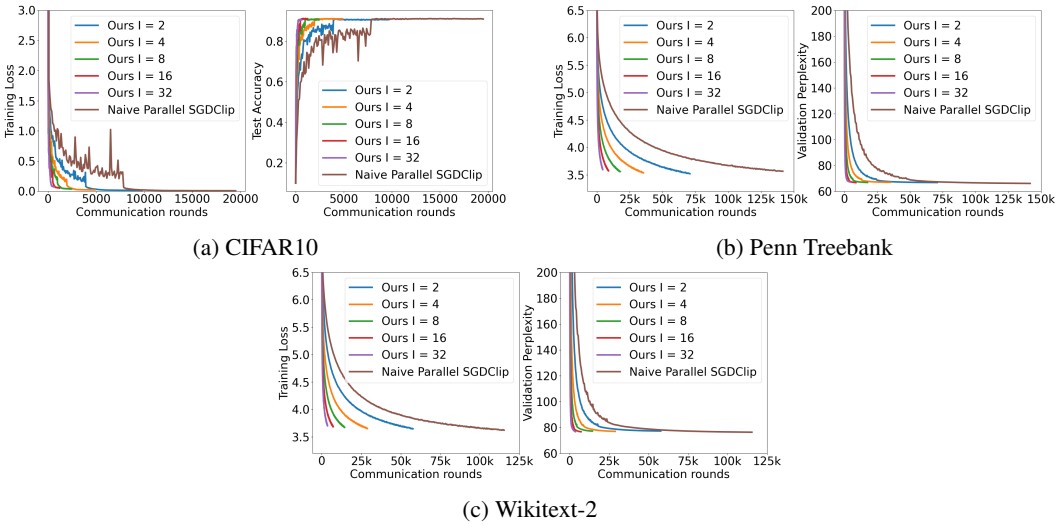

(a) CIFAR10              (b) Penn Treebank

(c) Wikitext-2

Figure 10: Performance v.s. # of communication rounds each GPU conducts on training (a) Resnet-56 on CIFAR10 (b) AWD-LSTM on Penn Treebank (c) AWD-LSTM on Wikitext-2.

### E.4   PERFORMANCE VS. COMMUNICATION ROUNDS

The reader might be interested in how the training and testing performance changes w.r.t. the number of communications each machine does for both the Naive Parallel SGDClip and our CELGC with different $I$. Thus, we show such results in Figure 10. The results are very close to that w.r.t. the wall-clock time which is as expected.

### E.5   PARTIAL PARTICIPATION

In the previous experiments, when communication between nodes occurs, all nodes send their model and/or gradients for averaging and update accordingly. Yet, in practice, it might be that only a (different) subset of nodes participate in communication each time. Therefore, we conduct an experiment on this setting. Specifically, we take the Penn Treebank dataset and train the AWD-LSTM of the same structure as in Section 5.1 to do language modeling. However, when the algorithm requires a communication across nodes, we randomly select 6 out of the total 8 nodes to communicate and average the model and/or gradient and update those 6 nodes only; the rest 2 nodes will be left untouched. Considering that this setting introduces big difference compared with the setting we adopted in Section 5.1, we finely tuned the initial learning rate for both the baseline and our algorithm and picked the one that gives the smallest training loss.

The results are reported in Figure 11. We can see that, though the partial participation setting slightly deteriorates the performance for both the baseline and our algorithm as compared with Figure 2a, our algorithm is still able to closely match the baseline for both the training loss and the validation

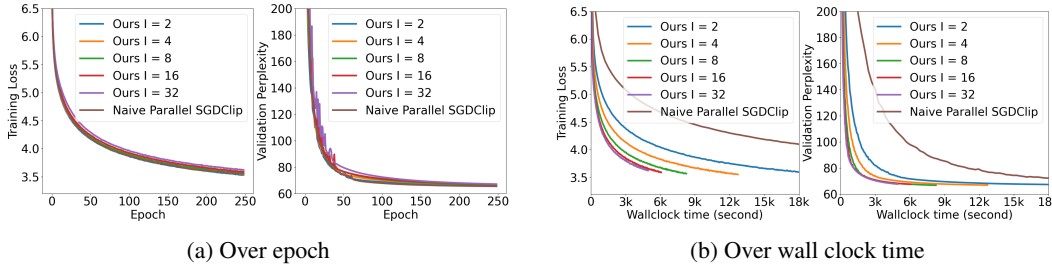

(a) Over epoch

(b) Over wall clock time

Figure 11: Train an AWD-LSTM to do language modeling on Penn Treebank where only a subset of nodes participate in each communication.

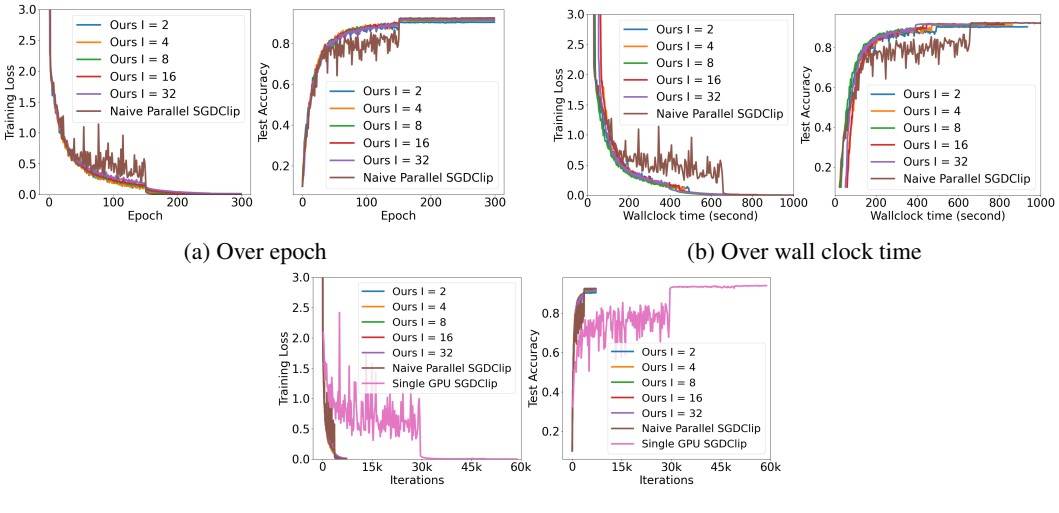

(a) Over epoch

(b) Over wall clock time

(c) Over # of iterations each GPU runs

Figure 12: Algorithm 1 with different $I$: Training loss and test accuracy v.s. (a) epoch, (b) wall clock time, and (c) # of iterations each GPU runs on training a 56 layer Resnet to do image classification on CIFAR10.

perplexity epoch-wise, but also note that our algorithm greatly saves wall-clock time. This indicates that our algorithm is robust in the partial participation setting.

### E.6 LARGE MINI-BATCH

The distributed learning paradigm is especially desirable with training using large mini-batches. To this end, we trained a Resnet-56 model on CIFAR10 with batch-size 256 on each node which sums up to a batch-size of 2048 globally. Compared with the experment in Section 5.1, considering the large batch-size, we trained for 300 epochs instead of 200, decrease the learning rate by a factor of 10 at epoch 150 and 250, and finely tuned the initial learning rate for each variant and picked the one that gives the smallest training loss. Results are shown in Figure 12. Similar to small batch case reported in Figure 1, our algorithm with skipped communication is able to match the naive parallel SGDClip in terms of epochs but greatly speeds up wall-clock-wise. Also, Figure 12c again verifies the parallel speed-up property of our algorithm.

