# OpenReview forum: "A Communication-Efficient Distributed Gradient Clipping Algorithm for Training Deep Neural Networks"
_ICLR.cc/2022/Conference — ICLR 2022 Submitted_

### Official Review · Reviewer_nQu7 · 2021-11-02

**Correctness:** 1
**Technical Novelty And Significance:** 2
**Empirical Novelty And Significance:** 2
**Recommendation:** 3
**Confidence:** 4

**Main Review:**

## Strengths
1. **Motivation and clarity.** The paper is well-motivated and easy to follow. The authors also provide a sketch of the proof in the main text which is useful.

2. **Related work** section provides a good summary of existing works relevant to the topic. However, Table 1 is not accurate and should be improved (see **General questions and comments** below).




## Weaknesses
1. **The proofs contain mathematical mistakes that break the main result of the paper.** In the proof of Lemma 2 the authors rely on the following formula (page 15, the third step at formula (12)): $\mathbb{E}\left[ \sum_{i\in \overline J(t)} \nabla F_i(x_t^i, \xi_t^i) | \xi^{[t-1]}\right] = \sum_{i\in \overline J(t)} \nabla f_i(x_t^i)$. This is not true since the set $\overline J(t)$ depends on the stochasticity at iteration $t$ of the algorithm. Moreover, the authors repeat this mistake several times throughout the proof when "move" the factor $|\overline J(t)|$ outside the expectations (e.g., later in the same formula and also when applying (12) to (9) in (14)). Due to the same issue, one cannot use second-moment representation in the fourth row of the formula (13). Finally, even if we assume that everything is correct before the last step in formula (13), there is another issue: in the last row of (13), the second term should be $2\frac{\eta^2 |\overline J(t)|^{\color{red}2}}{N^2}$. Therefore, in the final bound, instead of the term $\frac{AL_0\eta^2\sigma^2}{N}$ one should have $AL_0\eta^2\sigma^2$. As the result, in the last upper bound from the proof of Theorem 1 the third term should be $\frac{AL_0\eta^2\sigma^2}{\epsilon}$ instead of $\frac{AL_0\eta^2\sigma^2}{{\color{red}N}\epsilon}$. Therefore, the number of iterations $T$ is not proportional to $\frac{1}{N}$, i.e., **linear speed-up is not proven**. Since it is claimed as one of the main contributions of the paper, the described issue is a very strong reason for the rejection of the paper.

2. **Assumptions on the parameters in Theorem 1** are strong and not well-defined for some special cases. First of all, the condition $N \leq \frac{1}{\epsilon}$ is strong and in FL-applications, where $N$ can be large, it is typically satisfied for very small values of $\epsilon$, meaning that the problem should be solved with too high accuracy. In practice, it is often sufficient to solve the optimization problem with not too high accuracy. Next, the authors also assume that the number of local steps between two subsequent communications satisfies $I \leq \frac{1}{2N\epsilon}$. Since $I$ is a positive integer one should have $N \leq \frac{1}{2\epsilon}$. So, in fact, it means that either $\epsilon$ is extremely small, or $I = O(1)$, or $N$ is small. The drawbacks of the first and the third options are covered above. The second option describes a theoretically simple case since the number of local steps is small. Moreover, in this case, $N\epsilon = \Omega(1)$ meaning that the derived communication complexity bound is not better than for Naive Parallel SGD with clipping. Finally, the assumptions on $\gamma$ and $\eta$ are not well-defined when $\sigma = 0$ and $\kappa = 0$: the condition $\frac{\gamma}{\eta} = 5(\sigma + \kappa)$ implies that $\frac{\gamma}{\eta} = 0$ in this case meaning that either $\gamma = 0$ (no clipping) or $\eta = \infty$. This means that the assumptions on $\gamma$ and $\eta$ are incorrect.

3. **Numerical experiments** are conducted for the system with $N = 2$ workers, which is a too small number to demonstrate parallel speed-up properly. Moreover, it would be better to show the dependence on the number of communications rounds in the experiments. Moreover, to have a fair comparison of the method with local updates and the method without them (Naive Parallel SGDClip) one should use I times larger batchsizes for Naive Parallel SGDClip. It seems that the authors used the same batchsizes for local and non-local methods for each iteration.



## General questions and comments
1. **Page 2, "... and hence is communication-efficient".** This sentence should be rewritten since in the current form it implicitly says that local steps ensure communication efficiency. In fact, it is not true for highly heterogeneous or arbitrary heterogeneous cases (e.g., see Woodworth et al. (2020a) and Gorbunov et al. (2020)).

2. **Table 1** requires clarifications. First of all, Ghadimi & Lan (2013) do not use the bounded gradients assumption. They derive the results under classical smoothness and bounded variance assumptions. This should be explicitly stated either in the table or in the caption of the table. The complexity guarantee should contain two terms: one is proportional to $\epsilon^{-2}$ and the second one is proportional to $\epsilon^{-4}$. Moreover, their result can be easily generalized to the naive parallel settings without any assumptions on bounded data. Secondly, in the current form, the complexity bounds for the methods are incorrect for the special case when $\sigma = 0$ or/and $\kappa = 0$ since the terms with better dependence on $\epsilon$ are omitted. This should be fixed. Next, the table should contain the results for Local-SGD under the classical assumptions to have a clear comparison (e.g., from Koloskova, Anastasia, et al. "A unified theory of decentralized SGD with changing topology and local updates." International Conference on Machine Learning. PMLR, 2020). Finally, I have not found the rate presented in the row "Naive Parallel of (Zhang et al., 2020)" in (Zhang et al., 2020). This should be clarified (ideally, rigorous proof should be provided).

3. **Missing references.** I believe the authors should at least mention the work Reddi, Sashank, et al. "Adaptive federated optimization." arXiv preprint arXiv:2003.00295 (2020) since this paper considers very relevant methods such as Federated Adam.

4. **Assumption 1 and the remark after it.** Assumption (iii) is too restrictive: it is not satisfied for the noise with the unbounded domain such as Gaussian noise. The authors claim that "it is a normal assumption when encountering relaxed smoothness". However, this claim is not explained. I do not see any evidence of why this assumption is reasonable in this case. I understand that it is used in prior works, but I guess it is because of the difficulty of analyzing the methods under relaxed smoothness. Next, assumption (iv) is also quite restrictive. For example, in the convex case, one can analyze Local SGD and its variants without this assumption. I understand that in the non-convex case it is used in prior works, but it is better to write about this explicitly.

5. **Page 5, "... our algorithm is expected to have better performance."** This is not true when $\kappa$ is large: even Local SGD does not benefit from local steps in this case.

6. **Additional comments about Theorem 1.** Assuming that the formula for the communication complexity is correct, it is still not clear why it is better than the mentioned complexity of Naive Parallel SGD with clipping since $\kappa$ can be large.

7. **Page 6, "Another interesting fact is that both iteration complexity and communication complexity only depend on $L_0$."** Most likely it means that the authors just omitted some important dependencies in the complexity estimates.

8. **Page 18, lower bound for $T$.** The detailed derivation should be added after fixing all mistakes in the proof.


## Minor comments
1. **Lemma 2, "If $2\gamma I \leq c/L_1$ for some $c > 0$, then $\gamma \leq 2\gamma I \leq \frac{c}{L_1}$".** This is a trivial statement.

2. **Page 14, "For each iteration $t$, We"** $\longrightarrow$ "For each iteration $t$, we"

3. **Page 15, "and (b) holds due to Lemma 1 and Lemma 5"** $\longrightarrow$ "and (d) holds due to Lemma 1 and Lemma 5"

4. **Lemma 3 restated, page 16:** in the RHS one should have $\|\nabla f(\overline x_t)\|$

5. **Inequality (17), the second part**: dependencies on $\sigma + \kappa$ are missing.

6. **Page 18, definition of $V(x)$:** $\widetilde{A} \longrightarrow A$

**Summary Of The Paper:**

The paper proposes a new variant of SGD with clipping and infrequent communications (local updates) called Communication Efficient Local Gradient Clipping (CELGC) aimed at solving non-convex federated learning problems under the generalized smoothness ($(L_0, L_1)$-smoothness) assumption. The authors derive ergodic convergence guarantees for the convergence of CELGC to the first-order $\epsilon$-stationary point assuming additionally that the noise in stochastic gradients is bounded with probability $1$ and heterogeneity of the local data on clients is also bounded. Although the authors claim that their result shows that CELGC achieves linear speed-up and has better communication complexity than the naive version of parallel Clipped SGD, the proofs contain several significant inaccuracies making the main result of the paper incorrect in general. Moreover, several assumptions about the parameters such as the number of workers $N$ and the number of local steps between two consequent communication steps $I$ are restrictive.

**Summary Of The Review:**

To sum up, several parts of the proof should be corrected and the theory should be extended to support more general choices of $N$ and $I$. Moreover, several parts of the paper such as the comparison of the known complexity results with the derived ones and numerical experiments require improvements.

Unfortunately, the current version of the paper cannot be accepted to the conference. If the authors resolve the mentioned issues during the rebuttal, I will increase my score.

---

> ### Author Response · Authors · 2021-11-13
> **Reply [2/2]**
>
> 8. **Table 1 requires clarifications.**
> > A: We added comments for Ghadimi and Lan (2013)’s complexity results for Table 1 footnote.
> >
> > Due to limited space, we cannot add all local SGD results in Table 1. We added a red text after mentioning Table 1, to refer the readers to (Koloskova Anastasia, et al. ICML 2020) for local SGD complexity results for $L$-smooth functions.
> >
> > We added a description of the Naive Parallel of (Zhang et al. 2020) in Table 1 footnote. It is different from our algorithm (CELGC) with $I=1$. The difference is that the naive version requires averaging gradients across all machines to update the model while CELGC only updates the model using local gradients computed in that machine. This also means that in each iteration, the naive version clips the gradient based on the globally averaged gradient while ours is only based on the local gradient.
>
> 9. **Missing references of Reddi, Sshank, et al. “Adaptive federated optimization”.**
> > A: We have included it in our related work section.
>
> 10. **Assumption 1 and the remark about it.**
> > A: Assumption 1 (iii) is needed because we have to use it for analyzing relaxed smooth functions, and it is also used in previous works. This assumption can be relaxed to sub-Gaussian, and we believe we can establish high-probability results using union bound.
> >
> > Assumption (iv) is a standard assumption to describe data heterogeneity. Our analysis also works with $\kappa=0$.
>
> 11. **Page 5, "... our algorithm is expected to have better performance." This is not true when  $\kappa$ is large: even Local SGD does not benefit from local steps in this case.**
> > A: We respectfully disagree. Local SGD for nonconvex heterogeneous cases can also skip communication rounds and still achieve linear speedup. See for example Table 1 of (Yu et al. 2019).
> >
> > (Yu et al. 2019). On the Linear Speedup Analysis of Communication Efficient Momentum SGD for Distributed Non-Convex Optimization. ICML 2019.
>
> 12. **Assuming that the formula for the communication complexity is correct, it is still not clear why it is better than the mentioned complexity of Naive Parallel SGD with clipping since  $\kappa$ can be large.**
> > A: To the best of our knowledge, there is no analysis of Naive parallel SGD with clipping for nonconvex relaxed-smoothness conditions under heterogeneous settings. However, if $\kappa$ is large, it is clear that naive parallel SGD with clipping also suffers from large $\kappa$. In addition, please note that our algorithm improves the communication complexity in terms of $\epsilon$ as illustrated in Table 1, compared with naive parallel SGD with clipping.
>
> 13. **Page 6, "Another interesting fact is that both iteration complexity and communication complexity only depend on $L_0$." Most likely it means that the authors just omitted some important dependencies in the complexity estimates.**
> > A: We respectfully disagree. It is due to our analysis technique instead of missing dependencies. Please note that [Zhang et al. 2020]’s computational complexity only depends on $L_0$ as well. Please check Table 1 of [Zhang et al. 2020].
> >
> > [ref] Zhang et al. Improved Analysis of Clipping Algorithms for Non-convex Optimization. NeurIPS 2020.
>
> 14. **Page 18, lower bound for $T$. The detailed derivation should be added after fixing all mistakes in the proof.**
> > A: The complexity results do not change after fixing mistakes. Please check our proof. Thank you.
>
> 15. **Minor comments.**
> > A: Thank you for mentioning them. We have fixed them in the new version.

---

> > ### Comment · Reviewer_nQu7 · 2021-11-26
> > **Response to the rebuttal (Part 1/2)**
> >
> > I thank the authors for their replies and for their attempt to address my criticism. I have read other reviews, the rebuttal, and checked the updated version of the paper. However, a significant part of my criticism is still valid, and, in particular, the proofs still contain mistakes. I will respond point by point following the replies of the authors.
> >
> > **1. Mistakes in the proofs.** Unfortunately, the proofs still contain mathematical mistakes. Although the proof of Lemma 7 is correct, Lemma 8 is incorrect. First of all, it does not hold when $N = 1$, because for $N=1$ the lemma implies that $\mathbb{E}[A_i B_i] = \mathbb{E}[A_i]\mathbb{E}[B_i]$, which does not hold in general. This mistake appears because the authors use that $[\widetilde{T} = s] \in \mathcal{F}_s$. In fact, for all $s$ sigma-algebra $\mathcal{F}_s$ is not well-defined: sorting of $B_i$ implies that $B_1$ depends on $B_2, B_3, \ldots, B_N$ (and the same for others), so, we cannot use their independence once we sorted them.
> >
> > Next, in the new version of the proof, the authors have new terms proportional to $\mathbb{E}\left\|\sum_{i=1}^N p_i \nabla f_i(x_t^i)\right\|^2$ (in inequality (14)). To get rid of these terms, one needs to choose $\eta \leq \frac{1}{2AL_0 N\mathbb{E}[|\bar J(t)|]}$, if I am not mistaken. However, this assumption on $\eta$ is not reflected in the statement of the lemma and of the main result. This is highly important since it changes the complexity of the algorithm.
> >
> > **2. On the assumptions on $N$ and $I$.** I understand the response. However, even in the example provided by the authors $\epsilon = 0.06$, meaning that $N \leq 4$ if we take $N$ as $\frac{1}{\sqrt{\epsilon}}$. This is prohibitively small: in real-world parallel systems $N$ is usually of the order $10^3$ and in Federated Learning it might be much larger. Therefore, the mentioned limitation is significant.
> >
> > **3. On the assumptions on $\gamma$ and $\eta$.** This fact reduces the impact of the contribution. Complexity results should be meaningful for any reasonable choice of parameters in order to provide a clean comparison with existing works.
> >
> > **Points 4 and 5.** Thank you for the clarifications.
> >
> > **6. On batchsizes.** Let me clarify what I mean. As I wrote in the review, Naive Parallel SGDClip should have $I$ times larger batchsizes than CELGC to get a fair comparison. This is a standard approach to comparing Minibatch methods with Local methods (Woodworth et al, 2020b), i.e., between two neighboring communication rounds, the methods should use the same number of oracle calls. In the current experiments, this condition is not satisfied.
> >
> > **8. Table 1 requires clarifications.** The iteration complexity column contains inconsistency: for SGD the iteration complexity contains all terms, i.e., optimization ($\sigma = 0$) and stochastic terms ($\sigma > 0$), whereas the third and the fourth rows do not. Moreover, it should be clearly mentioned that in previous results Assumption 1 (iv) is not used. This is an important difference, but it is not reflected in the table.
> >
> > **9. Reference to Reddi, Sashank, et al., 2020.** In the current version of the paper, it is still not explained that they actually were the first who considered clipping-like methods (Adam can be viewed as an adaptive version of component-wise gradient clipping) with infrequent communication.
> >
> > **10. On Assumptions 1 (iii) and (iv).** More discussion of these assumptions should be added to the paper. In particular, it should be clearly stated that they are very strong. Regarding the relaxation of Assumption 1 (iii) to sub-Gaussian noise: although it might be done, I can only evaluate what is done in the paper.
> >
> > **11. On the efficiency of Local-SGD when $\kappa$ is large.** Linear speedup does not contradict my claim, because it will be achieved for very small $\epsilon$. What I am saying is that for large $\kappa$ the term depending on $\kappa$ will be leading for any reasonable choice of $\kappa$ when $I > 1$ (see [Koloskova, A., Loizou, N., Boreiri, S., Jaggi, M., & Stich, S. (2020, November). A unified theory of decentralized SGD with changing topology and local updates. In International Conference on Machine Learning (pp. 5381-5393). PMLR.]).
> >
> > **12. On the comparison with Naive Parallel SGDClip.** The rate of Naive Parallel SGDClip does not depend on $\kappa$. So, my criticism is still valid.
> >
> > **13. On the dependence on $L_1$.** The authors do not provide clarifications why their rates do not depend on $L_1$. This phenomenon deserves much more attention because one can choose $L_1$ to be extremely large making $L_0$ tiny for all points except the ones that are too close to the stationary points (where the norm of the gradient is too small). An intuitive explanation is required at least. Moreover, the proofs are not rigorous enough, because the central places are unexplained (see the next point).
> >
> > **14. Lower bound for $T$.** The authors have not updated this place (criticism is valid).

---

> > > ### Author Response · Authors · 2021-11-27
> > > **Thank you for the further feedback**
> > >
> > > We thank you for your efforts in giving your detailed further concerns which we address below:
> > >
> > > (1). **Mistakes in the proofs: restriction on $\eta$ and Lemma 8.**
> > > > A: We do not require such a restriction, since we have $AL_0\eta\leq 1/2$. This can guarantee that the negative coefficient can dominate the positive one in front of the term $E||\sum_{i=1}^{N}p_i\nabla f_i(x_t^i)||^2$, and hence can have an upper bound of zero. Please check Formula (14). The concrete formula of Lemma 8 is also not important, since they can cancel out due to the dominating coefficients (see Formula (14)). We will make it more clear in the revision.
> > >
> > > (2). **On the assumptions on $N$ and $I$.**
> > > > A: This criticism is not fair for such a theoretical work. We again want to emphasize that $\epsilon$ can be arbitrarily small.
> > >
> > > (3). **On the assumptions on $\gamma$ and $\eta$.**
> > > > A: Even the paper on SGDClip on a single machine requires that $\sigma > 1$ in their Theorem 3.2 [Zhang et al., 2020]. So we found our assumption also reasonable.
> > > >
> > > > [ref] Zhang et al. Improved Analysis of Clipping Algorithms for Non-convex Optimization. NeurIPS 2020.
> > >
> > > (6). **On batchsizes.**
> > > > A: The way in [Woodworth et al, 2020b] is certainly one way to compare, but we followed the setting of [Yu et al., 2019] which is also valid.
> > > >
> > > > [Yu et al. 2019]. On the Linear Speedup Analysis of Communication Efficient Momentum SGD for Distributed Non-Convex Optimization. ICML 2019.
> > >
> > > (8). **Table 1 requires clarifications.**
> > > > A: We will improve Table 1.
> > >
> > > (9). **Reference to Reddi, Sashank, et al., 2020.**
> > > > A: First, in their paper, they specifically pointed out that they “provide convergence analyses of these methods in general nonconvex settings, assuming full participation, i.e. $S = [m]$“.
> > > >
> > > > Second, even they themselves never claimed such a thing as clipping. Indeed, we could not follow your claim that “Adam can be viewed as an adaptive version of component-wise gradient clipping”? IMHO, Adam does something like normalization. Normalization downscales large values while upscaling small values so that they will become similar in magnitude after the operation; in contrast, clipping only cuts values that surpass a threshold while leaving those smaller than the threshold *untouched*. They are completely different things.
> > >
> > > (10). **On Assumptions 1 (iii) and (iv).**
> > > > A: We will add notes on these two, but we still want to emphasize that they have already been used in previous works and we find it too unfair to criticize us on this.
> > >
> > > (11). **On the efficiency of Local-SGD when $\kappa$ is large.**
> > > > A: We understand your claim but we only consider the case where $\epsilon \ll \kappa$.
> > >
> > > (12). **On the comparison with Naive Parallel SGDClip.**
> > > > A: We want to emphasize that $\epsilon$ can be arbitrarily small, while $\kappa$ is a constant once the setting of $N$ and the distribution of the data across machines are given.
> > >
> > > (13). **On the dependence on $L_1$.**
> > > > A: Once a problem is given, $L_0$ and $L_1$ will be fixed. They can be anything you want but they are fixed and independent of $\epsilon$. So no matter how large $L_1$ is, $\epsilon$ can be chosen to be small enough so that $\epsilon \ll 1/L_1$. Note that the complexity results in Table 1 in [Zhang et al., 2020] are also only dependent on $L_0$ since they also treat $\epsilon$ as arbitrarily small.
> > > >
> > > > [ref] Zhang et al. Improved Analysis of Clipping Algorithms for Non-convex Optimization. NeurIPS 2020.
> > >
> > > (14). **Lower bound for $T$.**
> > > > A: The complexity results do not change after fixing mistakes. Please check our proof. Thank you.
> > >
> > > (15). **19 page -> 24 pages.**
> > > > A: The major reason is that we added 3.5 pages of experiments that other reviewers asked for. We felt obliged to address their concerns as well and we didn’t realize that this would be an issue.

---

> > > > ### Comment · Reviewer_nQu7 · 2021-11-27
> > > > **Additional clarifications regarding my concerns**
> > > >
> > > > Unfortunately, the reply of the authors does not resolve the issues mentioned in my review and my response to the rebuttal. To be precise, I will reply point-by-point.
> > > >
> > > > **1.** To cancel the term $AL_0\eta^2 \left\|\mathbb{E}\left(\sum_{i=1}^N p_i \nabla f_i(x_t^i)\right)\right\|^2$ by $-\frac{\eta}{2}\frac{N}{\mathbb{E}|\bar{J}(t)|}\mathbb{E}\left\|\frac{1}{N}\sum_{i=1}^N p_i\nabla f_i(x_t^i)\right\|^2 = -\frac{\eta}{2}\frac{1}{N\mathbb{E}|\bar{J}(t)|}\mathbb{E}\left\|\sum_{i=1}^N p_i\nabla f_i(x_t^i)\right\|^2$ it is not sufficient to ensure that $AL_0\eta \leq 1/2$. The derivation in formula (14) implicitly uses that $AL_0\eta \leq \frac{1}{2N\mathbb{E}|\bar{J}(t)|}$. This should affect the final rate.
> > > >
> > > > Regarding the mistake in Lemma 8: unfortunately, the mistake is in the submission. I think the authors agree with this fact. As a reviewer, I should evaluate only the content of the submission. Therefore, unfortunately, I cannot take into account the authors' response that this place can be corrected (especially when the correction is not given).
> > > >
> > > > **2.** The criticism is fair even if we consider the paper as purely theoretical because state-of-the-art works on methods with infrequent communications (local steps) do not use such an assumption. Moreover, there is no explanation for why this assumption should be used. Taking into account what I wrote in my previous comment, I treat this assumption as a strong limitation.
> > > >
> > > > **3.** This argument does not address the essence of my criticism. I did not find a justification of why the lower bound for $\sigma$ is required. The fact that this was used in previous works does not give an answer to my question. However, the question is important for a better understanding of methods with clipping and their comparison with non-clipped methods. It is worth addressing this question in a single-node case first before moving to the distributed scenario.
> > > >
> > > > **6.** When the communication is a major bottleneck, the way of comparison from [Woodworth et al, 2020b] is better.
> > > >
> > > > **8.** Ok, but as I mentioned this table requires a major revision, therefore, an additional round of review is required.
> > > >
> > > > **9.** Regarding the first part: I did not get what the authors mean by quoting this claim. Both [Reddi et al., 2020] and the submission consider the full participation of clients. Regarding the second part: my claim is supported by [Zhang et al., 2019].
> > > >
> > > > **10.** Assumption 1 (iii) is not used in the analysis of Local-SGD. Assumption 1 (iv) is not used in the Naive Parallel SGDClip. The proposed method (CELGC) is based on both Assumption 1 (iii) and (iv). In some sense, the proposed analysis takes the strongest assumptions from both lines of works. As the result, we obtain a very restrictive set of assumptions that is stronger than in previous SOTA papers. This fact should be emphasized and a proper comparison should be added that takes into account all the differences between the assumptions.
> > > >
> > > > **11.** This should be explicitly mentioned in the text where it is needed.
> > > >
> > > > **12.** There are simple problems when $\kappa = +\infty$ on $\mathbb{R}^d$, e.g., minimization of the sum of different quadratic functions (with different Hessians). It can be also arbitrarily large. Therefore, the dependence on $\kappa$ is not a good property of the complexity bound. When one bound depends on $\kappa$ and another bound does not, one should be extremely careful with the comparison of these two bounds.
> > > >
> > > > **13.** In Theorem 1, the authors introduce the assumptions on $N$ and $\gamma$ that depend on $L_1$. Moreover, $\eta$ has an implicit dependence on $L_1$ as well since the authors make an assumption on a ratio between $\gamma$ and $\eta$. The final bound in the proof depends on $N$, $\eta$, and $\gamma$. Since the authors do not provide a complete derivation of the final result, it is complicated to infer all hidden dependencies on $L_1$. The authors should add complete proof. Referring to the previous work does not resolve the criticism because that work also may contain some hidden dependencies.
> > > >
> > > > **14.** As I explained, the proof is not complete (furthermore, it contains the mistakes explained in **1**) and the lower bound for $T$ is not properly derived. It is the authors' responsibility to provide complete proof. I pointed to the issue twice, but the authors have not addressed it.
> > > >
> > > > **15.** The significant increase of the submission length is a serious indicator that the paper requires an additional round of reviewing. Moreover, the authors significantly changed the proofs: almost 3 pages of the proofs are colored in red (red = changed part). It is almost half of all proofs. For a theoretical paper, it is a critical change.
> > > >
> > > > To sum up, the replies of the authors have not resolved the issues I mentioned. I believe that the only right decision now for AC is to reject the paper, and for the authors -- to address all the issues and to resubmit the paper to another conference/journal.

---

> > > > > ### Author Response · Authors · 2021-11-29
> > > > > **Reply (2/2)**
> > > > >
> > > > > (3). **This argument does not address the essence of my criticism. I did not find a justification of why the lower bound for $\sigma$ is required. The fact that this was used in previous works does not give an answer to my question. However, the question is important for a better understanding of methods with clipping and their comparison with non-clipped methods. It is worth addressing this question in a single-node case first before moving to the distributed scenario.**
> > > > >
> > > > > > A: Actually, we can change the value $1$ in the Assumption $\sigma + \kappa \ge 1$ to be any positive constant without changing the order of the complexity, and choosing it to be $1$ is nothing but a way to simplify the proof. Indeed, we only require that $\sigma + \kappa > 0$. The reason is that we did not attempt to design a universal algorithm that can automatically adapt to both the noiseless and the noisy settings.
> > > > >
> > > > > (6). **When the communication is a major bottleneck, the way of comparison from [Woodworth et al, 2020b] is better.**
> > > > >
> > > > > > A: We certainly acknowledge that the way of comparison from [Woodworth et al, 2020b] is valid, but we don’t find that rules out the validity of the way we used which was inherited from [Yu et al, ICML 2019]. And we were not sure what are the exact advantages of the former over the latter.
> > > > >
> > > > > (9). **Regarding the second part: my claim is supported by [Zhang et al., 2019].**
> > > > >
> > > > > > A: Sorry but we did not find which part of the paper supported your claim of “Adam can be viewed as an adaptive version of component-wise gradient clipping”.
> > > > >
> > > > > (10). **Assumption 1 (iii) is not used in the analysis of Local-SGD. Assumption 1 (iv) is not used in the Naive Parallel SGDClip. The proposed method (CELGC) is based on both Assumption 1 (iii) and (iv). In some sense, the proposed analysis takes the strongest assumptions from both lines of works. As the result, we obtain a very restrictive set of assumptions that is stronger than in previous SOTA papers. This fact should be emphasized and a proper comparison should be added that takes into account all the differences between the assumptions.**
> > > > >
> > > > > > A: Local-SGD does not use Assumption 1 (iii) since they have the $L$-smooth assumption while we do not have. We are considering a more challenging problem ($(L_0,L_1)$-relaxed smooth functions). This assumption is also used by [Zhang et al, 2019, Zhang et al. 2020].
> > > > > >
> > > > > > Assumption (iv) is a standard way to quantify heterogeneity of data across machines (e.g., [Yu et al. ICML 2019]). We are not aware of other weaker assumptions to characterize this in nonconvex settings. We will add more discussions in the revision.
> > > > >
> > > > > (12). **There are simple problems when $\kappa = \infty$ on $\text{R}^d$, e.g., minimization of the sum of different quadratic functions (with different Hessians). It can be also arbitrarily large. Therefore, the dependence on  is not a good property of the complexity bound. When one bound depends on  and another bound does not, one should be extremely careful with the comparison of these two bounds.**
> > > > >
> > > > > > A: The only result we know of that can handle your quadratic problem is the paper [Woodworth et al., 2020a] where they only consider the strongly convex and “weakly-convex” settings where each local objective is convex. Other previous work focusing on general smooth non-convex settings [e.g., Yu et al. 2019, Karimireddy et al., 2020, Koloskova et al., 2020] cannot handle your case with finite $\kappa$ (some work use other notations, such as $G$ in [Karimireddy et al., 2020] and $\hat{\zeta}$ in [Koloskova et al. 2020]). Please note that our setting is non-convex and relaxed smooth.
> > > > >
> > > > > (13). **In Theorem 1, the authors introduce the assumptions on $N$ and $\gamma$ that depend on $L_1$ ... Referring to the previous work does not resolve the criticism because that work also may contain some hidden dependencies.**
> > > > >
> > > > > > A: As we said, when $\epsilon$ is sufficiently small, the upper bound on $\gamma$ will be dominated by $\frac{N\epsilon^2}{28AL_0(\sigma + \kappa)}$ which means it does not depend on $L_1$, and in turn $\eta$ will not depend on $L_1$ as well.
> > > > > Then, from the full formula of the upper bound in Appendix B.4 the last inequality, we can see the only dependence on $L_1$ comes from $N$ which we already included in Theorem 1.
> > > > >
> > > > >
> > > > > (15). **The significant increase of the submission length is a serious indicator that the paper requires an additional round of reviewing. Moreover, the authors significantly changed the proofs: almost 3 pages of the proofs are colored in red (red = changed part). It is almost half of all proofs. For a theoretical paper, it is a critical change.**
> > > > >
> > > > > > A: The main changes are to change the definition of $\text{E}\left[\nabla F_i(x_t^i;\xi_t^i)\mathbb{I}(i\in \bar{J}(t))\right]$ due to the previous issue you mentioned, and the rest analysis is almost unchanged. However, we have to change everywhere so we marked all of them in red. We do not think this is a lot of change.

---

> > > > > > ### Comment · Reviewer_nQu7 · 2021-11-29
> > > > > > **Thanks for the reply**
> > > > > >
> > > > > > I thank the authors for the detailed replies to my criticism. I acknowledge their efforts and time to address my concerns.
> > > > > >
> > > > > > However, during the discussion with the authors, a number of places were indicated that should be fixed. To make a justified decision on the paper, it is crucial to see the final version of the paper that incorporates all necessary changes (including modified proofs and assumptions on the noise) and addresses the mentioned issues. Therefore, I am still convinced that another round of review is required for the paper, and, as a result, I cannot recommend acceptance of the paper. I believe extra work on the submission and an additional round of reviews will only benefit the paper. I strongly encourage the authors to address all the issues pointed by the reviewers and resubmit the paper to another conference or a journal. ICML or JMLR might be a good fit for the paper.

---

> > > > > ### Author Response · Authors · 2021-11-29
> > > > > **Reply [1/2]**
> > > > >
> > > > > (1). **Cancel the term …**
> > > > >
> > > > > > A: Thanks for pointing it out. It is a typo instead of an error. Please see the formula (b) on page 17. The term $\|E(\sum_{i=1}^{N}p_i\nabla f_i(x_t^i))\|^2$ should be $\|E(\textcolor{red}{\frac{1}{N}}\sum_{i=1}^{N}p_i\nabla f_i(x_t^i))\|^2$, so the cancellation in (14) works when $AL_0\eta\leq 1/2$. We will fix it.
> > > > >
> > > > >  **Regarding the mistake in Lemma 8.**
> > > > >
> > > > > > A: We acknowledge Lemma 8 does not hold in general. We need some further assumptions: **if the stochastic gradient’s noise has symmetric distribution around its mean and the density is monotonically decreasing over the distance between the mean and the random variable (which is empirically verified in Figure 1(b) of [Zhang et al. NeurIPS 2020])**, then we can prove that $E(g\mathbb{I}(\|g\|\leq \alpha))=pE(g)$ for some $0\le p \le \text{Pr}(\|g\|\leq \alpha)$, where $g$ is the stochastic gradient and $\alpha$ is the clipping threshold. This mimics the conclusion of Lemma 8 and the rest of the proof can also go through since our formula (12) only requires an upper bound of $p_i$ (e.g., $p_i\leq \text{Pr}(\|g_i\|\leq \alpha)$). We will add more discussions in the revision.
> > > > > >
> > > > > > To give an illustration of the proof, we will take the $1$-dimensional case. This can be decomposed into $3$ scenarios depending on the mean $\mu$:
> > > > > >
> > > > > > i. $\mu = 0$, then obviously $E[g\mathbb{I}(\|g\|\leq \alpha)] = 0 = p * \mu$ where $0\le p\le \text{Pr}(\|g\|\leq \alpha)$.
> > > > > >
> > > > > > ii. $\mu > 0$. We first prove that $E[g\mathbb{I}(\|g\|\leq \alpha)]$ is also positive: $E[g\mathbb{I}(\|g\|\leq \alpha)] = \int_{-\alpha}^{0} gP(g)dg + \int_{0}^{\alpha} gP(g)dg = A_1 + A_2$. Now substitute $g$ in $A_1$ with $\hat{g} \triangleq -g$ we have $A_1 =  \int_{0}^{\alpha} -\hat{g}P(-\hat{g})d\hat{g}$. As $P(g) > P(-g)$ for $g > 0$ due to the monotonic decreasing of the probability density. Then $A_1 + A_2 > 0$.
> > > > > >
> > > > > > Next we prove that $E[g\mathbb{I}(\|g\|\leq \alpha)] \le \mu *  \text{Pr}[\|g\|\leq \alpha]$, this can again be decomposed into two cases:
> > > > > >
> > > > > > ii. a. $\alpha \le \mu$. Then $E[g\mathbb{I}(\|g\|\leq \alpha)]  \le  \mu * E[\mathbb{I}(\|g\|\leq \alpha)] = \mu * \text{Pr}[\|g\|\leq \alpha]$ as $g \le \mu$.
> > > > > >
> > > > > > ii. b. $\alpha > \mu$.
> > > > > >
> > > > > > \begin{equation}
> > > > > > E[g\mathbb{I}(\|g\|\leq \alpha)] = \int_{-\alpha}^{2\mu - \alpha}gP(g)dg + \int_{2\mu - \alpha}^{\mu}gP(g)dg + \int_{\mu}^{\alpha}gP(g)dg \triangleq A_1 + A_2 + A_3.
> > > > > > \end{equation}
> > > > > >
> > > > > > Now we substitute $g$ in $A_2$ with $\hat{g} \triangleq 2\mu - g$ then $A_2$ becomes $\int_{\mu}^{\alpha}(2\mu - \hat{g})P(2\mu - \hat{g})d\hat{g} = \int_{\mu}^{\alpha}(2\mu - \hat{g})P(\hat{g})d\hat{g}$ due to the symmetry around $\mu$ of the probability density function.
> > > > > Thus, $A_2 + A_3 = 2\mu * \int_{\mu}^{\alpha}P(g)dg = \mu * (\int_{2\mu - \alpha}^{\mu}P(g)dg + \int_{\mu}^{\alpha}P(g)dg)$ again due to the symmetry.
> > > > > >
> > > > > > Consequently,
> > > > > >
> > > > > > \begin{equation}
> > > > > > E[g\mathbb{I}(\|g\|\leq \alpha)] = A_1 + A_2 + A_3 \le \mu * \int_{-\alpha}^{\alpha}P(g)dg = \mu * \text{Pr}[\|g\|\leq \alpha].
> > > > > > \end{equation}
> > > > > >
> > > > > > iii. $\mu < 0$. This case can be proven similarly as $\mu > 0$.
> > > > > >
> > > > > > In conclusion, we have proved that $E(g\mathbb{I}(\|g\|\leq \alpha))=pE(g)$ for some $0\le p\le \text{Pr}(\|g\|\leq \alpha)$
> > > > > >
> > > > > > [Zhang et al., NeurIPS 2020] Zhang, J., Karimireddy, S. P., Veit, A., Kim, S., Reddi, S., Kumar, S., & Sra, S. (2020). Why are Adaptive Methods Good for Attention Models?. Advances in Neural Information Processing Systems, 33.
> > > > >
> > > > > (2). **The criticism is fair even if we consider the paper as purely theoretical because state-of-the-art works on methods with infrequent communications (local steps) do not use such an assumption. Moreover, there is no explanation for why this assumption should be used. Taking into account what I wrote in my previous comment, I treat this assumption as a strong limitation.**
> > > > > > A: First of all, we consider a *relaxed* $(L_0, L_1)$ smooth condition, unlike the $L$ smooth condition considered in previous works. Our setting is fundamentally more difficult.
> > > > > >
> > > > > > Second, this is actually implicitly assumed in [Yu et al., ICML 2019]. See Corollary 1 at Page 5. Note that $N$ is fixed once the setting is given, then as [Yu et al., 2019] themselves said (just above the Corollary statement), that when $T$ is sufficiently large, $O(\frac{1}{\sqrt{NT}})$ dominates $O(\frac{N}{T})$ which means $T \ge O(N^3)$. Then from $O(\frac{1}{\sqrt{NT}}) \le \epsilon^2$ we also have that $T \ge O(\frac{1}{N\epsilon^4})$. Finally, since $\epsilon$ can be arbitrarily small while $N$ is fixed we will have that $O(\frac{1}{N\epsilon^4}) \ge O(N^3)$ which gives us $N \le O(\frac{1}{\epsilon})$.
> > > > > >
> > > > > > Third, according to our Theorem 1, if there is no upper bound for $N$, then $\gamma$ can also be arbitrarily large. In turn, since $\gamma / \eta$ is a fixed constant, then $\eta$ can also be arbitrarily large. Then, these two parameters both being too large would lead to divergence.

---

> > ### Comment · Reviewer_nQu7 · 2021-11-26
> > **Response to the rebuttal (Part 2/2)**
> >
> > As I mentioned in Part 1 of my response, the main criticism is still valid. Unfortunately, the paper requires a major revision.
> >
> > I would like to emphasize as well that the initial version of the paper contained 19 pages. The current version has 24 pages meaning that more than 20% of the current version is new. However, the paper still contains mathematical mistakes and unexplained places in the central result. All these facts clearly indicate that the paper should be properly revised and requires another full round of reviewing. That is, the paper cannot be accepted in the current shape.
> >
> > In the revision for future submissions for the conferences/journals, I encourage the authors to put a lot of attention to the weaknesses indicated by the reviewers, in particular, to detailed explanation and comparison with the related work (Table 1 should be improved), to fixing the mistakes in the proofs, and to making the proofs complete (without unexplained derivations).

---

> ### Author Response · Authors · 2021-11-13
> **Reply [1/2]**
>
> We thank the reviewer for your careful review and constructive feedback, we would like to address your concerns below:
> 1. **The proofs contain mathematical mistakes that break the main result of the paper. $\bar{J}(t)$ depends on the stochasticity at iteration $t$ of the algorithm.**
> > A: Thank you very much for catching this. This was indeed an error in our proof.
> >
> > In our new version, we introduce two Lemmas (Lemma 7 and Lemma 8) in Appendix D to fix this issue.
> >
> > Lemma 7 is used to show that the clipped stochastic gradient has a smaller variance than the original stochastic gradient.
> >
> > Lemma 8 is used to decouple the dependency among the random variable $\bar{J}(t)$ and stochastic gradients at $t$-th iteration.
> >
> > Please also check the fixed proof on pages 15 and 16.
> >
> > Fortunately, it does not break our results of computation and communication complexity.
>
> 2. **Moreover, several assumptions about the parameters such as the number of workers and the number of local steps between two consequent communication steps are restrictive. It means that either $\epsilon$ is small, or $I=O(1)$ or $N$ is small.**
> > A: Thanks a lot for pointing this out.
> >
> > We first want to argue that $\epsilon$ is indeed small in practice. For example, we have drawn the plot of the $\ell_2$ norm of the gradient over the epoch curve for training the Resnet model on CIFAR10 (see Figure 8 in Appendix E.2). The results show that even for our setting, we can reach an $\epsilon$ of $0.6$ which means $N$ could be up to $16$ and thus our experimental setting is consistent with our theory.
> >
> > Second, $N$ and $I$ can be both moderately large as long as their product does not exceed $1/\epsilon$. For example, $N=1/\sqrt{\epsilon}$, $I=O(1/\sqrt{\epsilon})$ can also work. Please note that as long as $N<1/\epsilon$, our algorithm is proven to enjoy better communication complexity (please check the last two rows of Table 1).
> >
> > Consequently, these three options can be satisfied at the same time.
>
> 3. **The assumptions on $\gamma$ and $\eta$ are not well-defined when $\sigma=0$ and $\kappa=0$.**
> > A: Thank you for pointing it out.
> >
> > Actually, in the proof of Theorem 1, we require $\sigma+\kappa\geq 1$. We explicitly mention it in the statement of Theorem 1 in the new version. Please note that even for a single machine setting, the algorithm is not universal for deterministic and stochastic cases. The parameter $\gamma$ and $\eta$ need to be set very differently in these two cases. Please check Theorem 3.1 and Theorem 3.2 in [ref].
> >
> > [ref] Zhang et al. Improved Analysis of Clipping Algorithms for Non-convex Optimization. NeurIPS 2020.
>
> 4. **Numerical experiments are conducted for the system with $N=2$ workers, which is a too small number to demonstrate parallel speed-up properly.**
> > A: Sorry if we did not make it clear: we did all our experiments on $2$ machines each having $4$ V100 GPU cards and each GPU card is regarded as an individual node. Therefore, in our settings, $N = 8$. We consider this to be a decent number to demonstrate parallel speed-up.
>
> 5. **Moreover, it would be better to show the dependence on the number of communications rounds in the experiments.**
> > A: Thank you for pointing this out, we have added such plots in Appendix E.4.
>
> 6. **Moreover, to have a fair comparison of the method with local updates and the method without them (Naive Parallel SGDClip) one should use I times larger batchsizes for Naive Parallel SGDClip. It seems that the authors used the same batchsizes for local and non-local methods for each iteration.**
> > A: We are confused by this question: did you mean $N$ times larger batch sizes? If so, we respectfully disagree that using the same batch size for Naive Parallel SGDClip and our CELGC is unfair. The reason is that for both algorithms, each machine uses the same batch size which means globally, in each iteration, both algorithms access the same amount of data globally.
>
> 7. **​​Page 2, "... and hence is communication-efficient". This sentence should be rewritten.**
> > A: We deleted this sentence in the new version.

---

### Official Review · Reviewer_63cQ · 2021-11-02

**Correctness:** 4
**Technical Novelty And Significance:** 2
**Empirical Novelty And Significance:** 2
**Recommendation:** 5
**Confidence:** 4

**Main Review:**

The proposed CELGC is simple yet effective. The only difference from local sgd is that CELGC adopts normalized gradient if the norm of gradient is large. The authors also prove its convergence rate for relaxed smooth and non-convex functions.

In addition to convergence theorem, I care more about the practicality of the proposed algorithm and I have the following questions:
1. CELGC has many hyper-parameters, e.g., learning rate, clipping threshold and batch size. I notice that in the three experiments, these hyper-parameters are quite different. How to tune these hyper-parameters in a new training problems if we do not have any previous experience (In the experiments, authors often set parameters according to other papers). Is it possible to give an approximate range for these parameters?
2. The authors claim that CELGC can achieve the linear speedup with proper setting. But they seems do not conduct speedup experiment.
3. What the difference between the naive version of the parallel gradient clipping algorithm and CELGC with i=1?. I think they are the same one, i.e. \kappa = 0, so that the performance of the baseline should be better than CELGC with I>1 in the comparison of epochs. This is not consistent with Figure1(a). Besides, I think the authors should set a large batch size for the baseline in the experiments. Normalized gradients may lead to a bad performance when the batch size is small [1]

[1] Beyond convexity- Stochastic quasi-convex optimization. NIPS, 2015.

**Summary Of The Paper:**

This paper propose a novel distributed optimization method CELGC. CELGC adopts normalized gradient for local training (e.g., FL) so that it achieves better performance than SGD.

**Summary Of The Review:**

This paper has many things to be improved.

---

> ### Author Response · Authors · 2021-11-13
> **Reply**
>
> We thank the reviewer for liking our paper and would like to address your concerns below:
> 1. **CELGC has many hyper-parameters, e.g., learning rate, clipping threshold and batch size. I notice that in the three experiments, these hyper-parameters are quite different. How to tune these hyper-parameters in a new training problems if we do not have any previous experience (In the experiments, authors often set parameters according to other papers). Is it possible to give an approximate range for these parameters?**
> > A: We would like to emphasize that our algorithm does not introduce any additional hyper-parameter that needs to be tuned, compared with the single GPU version SGD clipping algorithm.
> >
> > From our experience during conducting the experiments, we give the following approximate ranges of these parameters:
> >
> > * CIFAR10 with resnets: initial learning rate $[0.01 \sim 1]$, clipping threshold $[1 \sim 10]$, batch size $[16 \sim 256]$
> >
> > * PennTreebank/Wikitext-2 with AWD-LSTMs: initial learning rate $[5 \sim 40]$, clipping threshold $[1 \sim 10]$, batch size $[3 \sim 20]$.
>
> 2. **The authors claim that CELGC can achieve the linear speedup with proper setting. But they seems do not conduct speedup experiment.**
> > A: We have already reported the speedup of training a Resnet model to do image classification in CIFAR10 (see Figure 4).
> >
> > In addition, we have also added Figure 7 in Appendix E.1 to show the speedup effect of our experiments compared with the single GPU version on training AWD-LSTMs to do language modeling on PennTreebank and Wikitext-2.
> >
> > We hope this suffices to convince the reviewer about the speedup effect of our algorithm.
>
> 3. **What the difference between the naive version of the parallel gradient clipping algorithm and CELGC with i=1?. I think they are the same one, i.e. \kappa = 0, so that the performance of the baseline should be better than CELGC with I>1 in the comparison of epochs. This is not consistent with Figure1(a).**
> > A: They are indeed different: the difference between the naive version of the parallel gradient clipping algorithm and CELGC with $I = 1$ is that the naive version requires averaging gradients across all machines to update the model while CELGC only updates the model using local gradients computed in that machine. This also means that in each iteration, the naive version clips the gradient based on the globally averaged gradient while ours is only based on the local gradient.
> >
> > Therefore, it is plausible that the baseline version could be no better than CELGC with $I > 1$ as reported in Figure 1(a).
>
> 4. **Besides, I think the authors should set a large batch size for the baseline in the experiments. Normalized gradients may lead to a bad performance when the batch size is small [1].**
> > A: We are working on it and will report the results once we get them.
> >
> > Meanwhile, we want to mention that [1] focuses on optimizing quasi-convex functions which is very different from us, since we consider non-convex relaxed-smooth functions. Actually, in Theorem 5.1 of [1], the authors require the minibatch size to be large (i.e., $\Omega(1/\epsilon^2))$ in their setting, while in our setting, constant minibatch can also converge. Please check our Theorem 1.
> >
> > [1]. Beyond Convexity: Stochastic Quasi-Convex Optimization. NIPS 2015.

---

> ### Author Response · Authors · 2021-11-18
> **Experiments added: Large Batch**
>
> We have updated the pdf with an experiment of large batch sizes: 256 for each machine which sums up to 2048 globally on training the Resnet-56 model on CIFAR10 which can be found in Appendix E.6 and Figure 12. It shows that the algorithm with skipped communication is able to match the naive version of SGDClip in terms of epochs but gains a speed up in terms of wall-clock time, which indicates that our algorithm is robust in the large batch setting.

---

### Official Review · Reviewer_oVjB · 2021-11-08

**Correctness:** 3
**Technical Novelty And Significance:** 2
**Empirical Novelty And Significance:** 2
**Recommendation:** 5
**Confidence:** 3

**Main Review:**

Gradient clipping has become a de facto approach in training the deep neural networks especially for LSTM and Transformer. When communication bandwidth is limited and sharing gradient can leak privacy in federated learning, global gradient clipping is impractical. In this case, practitioners use local gradient clipping as workaround. However, there is no theoretical understanding on this approach, This paper bridges the gap between theory and practice. Overall, I think the paper is well written and easy to understand. But the contribution is incremental. In addition, I have a few comments regarding theory and experiments:

1. Theorem 1 holds when $N \leq O(1/\epsilon)$ and $I \leq O(1/(\epsilon N))$. On one hand, if we need to use many workers such as $N = 1/\epsilon$, then $I \leq O(1)$, which means there is no local iterations at all. On the other hand, if we just need a rough solution (e.g., a moderate $\epsilon$), which is typically the case in deep learning, then $N$ has to be small. It seems to me the theory does not apply to common cases in practice. It is good to plot the norm of the gradient to see how many cases in the experiments satisfy the assumption.

2. All the experiments focus on homogeneous local data in classic distributed settings. There is no experiment considering federated learning settings where the local data is heterogeneous and only a subset of clients participate in each training round. For the classic distributed training, global gradient clipping is applicable because of high-speed InfiniBand on the cloud and fast NCCL implementation.


**Summary Of The Paper:**

Gradient clipping is an important technique in training deep neural network. Typically, ones need to use the globally averaged gradient to estimate the norm. However, it requires gradient synchronization for every iterations, which is not practical in federated learning. In practice, practitioners only apply local gradient clipping to the local iterates while the theoretical analysis is lacking. This paper analyzes the convergence of the local gradient clipping. The theoretical results show that the convergence is guaranteed when both the number of workers and the number of local iterations are not too large.

**Summary Of The Review:**

The paper provides theoretical analysis for local gradient clipping under the federated learning settings. However, the practical cases where the theoretical analysis can be applied is unclear. And, the experiments are based on homogeneous local data in classic distributed settings. The contribution is also incremental.

---

> ### Author Response · Authors · 2021-11-13
> **Reply**
>
> We thank the reviewer for the constructive feedback and would like to address your concerns below:
> 1. **the contribution is incremental**
> > A: We respectfully disagree. To the best of knowledge, our work is novel due to the following reasons:
> >
> > First, our design of the local clipping algorithm is new in Federated Learning. Indeed, as Reviewer qwWt wrote: “gradient clipping is known to be an effective tool in training RNNs... there is not much work on analyzing its effect in distributed or federated learning scenarios. This paper is among the few ones...”
> >
> > Second, our analysis is novel, and significantly different from single-machine proof as well as the traditional local SGD proof for $L$-smooth functions. Actually, we obtained our results under a relaxed version of the $L$-smoothness property which has been observed to be applicable to models like LSTM. In addition, we need to introduce novel measures to monitor algorithmic progress in FL settings since we do not have access to the measures used in the standard analysis. Please refer to Section 4.3 for details. In addition, we also introduce two new lemmas (Lemma 7 and Lemma 8), which are important to decouple the dependency among random variables due to the gradient clipping operation. These Lemmas are of independent interest and could inspire future research on clipping algorithms in distributed settings.
> >
> > Therefore, we invite the reviewer to reconsider her/his evaluation of our submission.
> 2. **Theorem 1 holds when $N\leq O(1/\epsilon)$  and $I\leq O(1/(\epsilon N))$ . On one hand, if we need to use many workers such as $N=1/\epsilon$ , then $I\leq O(1)$, which means there is no local iterations at all. On the other hand, if we just need a rough solution (e.g., a moderate ), which is typically the case in deep learning, then has to be small. It seems to me the theory does not apply to common cases in practice. It is good to plot the norm of the gradient to see how many cases in the experiments satisfy the assumption.**
> > A: Thanks a lot for pointing this out. When $N<1/\epsilon$ (e.g., $N=1/\sqrt{\epsilon}$), then $I$ is not $O(1)$ anymore (e.g., $I=O(1/\sqrt{\epsilon})$), and then it saves communication complexity in this case (please check the last two rows of Table 1).
> >
> > In addition, we have drawn the plot of the $\ell_2$ norm of the gradient over the epoch curve for training the Resnet model on CIFAR10 (see Figure 8 in Appendix E.2). The results show that for this setting, we can reach an $\epsilon$ of $0.06$ which means $N$ could be up to $16$ and thus our experimental setting is consistent with our theory.
>
> 3. **All the experiments focus on homogeneous local data in classic distributed settings. There is no experiment considering federated learning settings where the local data is heterogeneous and only a subset of clients participate in each training round. For the classic distributed training, global gradient clipping is applicable because of high-speed InfiniBand on the cloud and fast NCCL implementation.**
> > A: We have added some results on the heterogeneous setting (see Appendix E.3) which shows a consistent pattern with our homogeneous ones. Due to the time limit, we haven’t finished all $I$ values but we will update the complete picture once it’s available.
> >
> > For the local participation problem, we are still waiting for the results and will upload them once finished.

---

> ### Author Response · Authors · 2021-11-18
> **Experiment added: heterogeneous data distribution and a subset of clients participation**
>
> We have updated the pdf with experiment results under the setting of heterogeneous data distribution in which each machine only accesses a different subset of the whole dataset (Appendix E.3 and Figure 9) and the setting of only a subset of clients participating in each training round (Appendix E.5 and Figure 11). The results show that our algorithm can still match the baseline in terms of epochs but significantly speeds up in terms of wall-clock time. Thus, our algorithm is robust to both the heterogeneous and the partial participation settings. Thanks.

---

### Official Review · Reviewer_qwWt · 2021-11-08

**Correctness:** 4
**Technical Novelty And Significance:** 2
**Empirical Novelty And Significance:** 2
**Recommendation:** 6
**Confidence:** 4

**Main Review:**

Gradient clipping is known to be an effective tool in training RNNs and mitigating the gradient explosion. However, there is not much work on analyzing its effect in distributed or federated learning scenarios. This paper is among the few ones that tries to theoretically analyze the behavior of FedAvg and similar algorithms when the workers use gradient clipping in their local SGD updates.
The paper is generally well-written and the claims are well-supported. One missing recent paper in this area (which was published recently and I don't expect the authors to be aware of it) is the "Understanding Clipping for Federated Learning ..." by Xinwei Zhang, et. al., published recently at ICML'21 Workshop on Federated Learning. It would be nice if the authors can comment and compare their results with the findings in that paper and other similar works.

Some minor suggestions:
 - One nice addition to the paper might be to show empirically how tight the theoretical bounds are (at least for a toy example).
 - Assumption 1, iii, $\\nabla$ is missing in the first equation.
 - The paragraph after Lemma 2, last sentence, the error is quadratic in $I$, not linear.

**Summary Of The Paper:**

The paper considers the effect of gradient clipping in Federated Learning and how it affects the convergence rate. They focused on the relaxed-smooth loss function. Each worker uses local gradient clipping and runs multiple steps of SGD before communicating and averaging the local models. The authors theoretically analyzed the algorithm and showed that for $N$ workers, the algorithm has $O(1/N\\epsilon^4)$ iteration complexity to find an $\\epsilon$-stationary point. Finally, the theoretical results are experimentally verified on CIFAR-10 (Resent-56 model), Penn Treebank and WikiText (LSTM models).

**Summary Of The Review:**

The paper has theoretically analyzed the effect of gradient clipping in local SGD updates in Federated Learning and FedAvg. To the best of my knowledge, the technical analysis and results are incrementally novel and improves the results of existing works.

---

> ### Author Response · Authors · 2021-11-13
> **Reply**
>
> We thank the reviewer for liking our paper and would like to address your concerns below:
> 1. **Compare with https://arxiv.org/abs/2106.13673**
> > A: Thank you for the information, we have cited this paper and mentioned it in the related work section. Both that paper and ours are investigating the usage of clipping in the federated setting. Nevertheless, they considered employing gradient clipping to optimize the $L$-smooth function and achieve differential privacy; in contrast, our focus is on designing algorithms to optimize relaxed-smooth function in the Federated Learning setting with provably linear speedup.
> 2. **Assumption 1, iii, $\nabla$ is missing in the first equation.**
> > A: Thank you for your careful review. We have fixed it.
> 3. **The paragraph after Lemma 2, last sentence, the error is quadratic in I, not linear.**
> > A: It is indeed linear as the paragraph after Lemma 2 is to describe Lemma 3.

---

### Author Response · Authors · 2021-11-13
**Thank you for your constructive feedback, we have revised the paper to reflect the changes.**

We sincerely thank all reviewers for their constructive feedback. We have fixed all mistakes in proofs and done some of the experiments which are all added to the appendix. All major changes are marked in red color. We are still running the other experiments reviewers suggested and will update once we get the results.

---

### Author Response · Authors · 2021-11-22
**Feedback request**

Dear reviewers:

We truly appreciate your constructive feedback and the efforts you put into reviewing our work. We believe we have addressed all your concerns in our responses and the revised paper.

First, we have fixed all mistakes in the proof.

Second, we have conducted several experiments in various scenarios according to your suggestions, specifically:
1. The heterogeneous setting in which each machine only accesses a different subset of the whole dataset (Appendix E.3).
2. The partial participation setting in which only a different subset of machines participate in each communication round and get updated (Appendix E.5).
3. The large mini-batch setting in which, for the CIFAR10 experiment, we use 256 batchsize for each machine that sum up to 2048 batch size globally (Appendix E.6).

In all settings, our algorithm is able to match the naive version of SGDClip in terms of epochs but greatly speeds up in terms of wall-clock time.

We hope the reviewers could take these new results into consideration and re-evaluate our work.

---

### Decision · Program_Chairs · 2022-01-20

**Decision:**

Reject

**Comment:**

This paper made a solid contribution studying the convergence rate of a simple distributed gradient clipping algorithm. The proposed algorithm simply clips the gradients on each local machine and then do simple distributed update of the parameters.

The result, if correct, is quite strong and significant: The proposed algorithm is simple, and shows some benefit comparing to previously proposed algorithms -- The strongest part of the paper is that it comes with a convergence rate bound (which is typically hard to prove for gradient clipping methods).


However, during the rebuttal period it was discovered that a number of places in the proofs are not well-supported, the paper has to go through major revision in order to meet the publication standard.